# No-regret Algorithms for Fair Resource Allocation

**Abhishek Sinha, Ativ Joshi**
School of Technology and Computer Science
Tata Institute of Fundamental Research
Mumbai 400005, India
abhishek.sinha@tifr.res.in
ativ@cmi.ac.in

**Rajarshi Bhattacharjee, Cameron Musco,**
**Mohammad Hajiesmaili**
Manning College of Information
and Computer Sciences
University of Massachusetts Amherst
{rbhattacharj, cmusco,
hajiesmaili}@cs.umass.edu

## Abstract

We consider a fair resource allocation problem in the no-regret setting against an unrestricted adversary. The objective is to allocate resources equitably among several agents in an online fashion so that the difference of the aggregate $\alpha$-fair utilities of the agents achieved by an optimal static clairvoyant allocation and the online policy grows sublinearly with time. The problem inherits its difficulty from the non-separable nature of the global $\alpha$-fairness function. Previously, it was shown that no online policy could achieve a sublinear standard regret in this problem. In this paper, we propose an efficient online resource allocation policy, called Online Fair Allocation (`OFA`), that achieves sublinear $c_\alpha$-approximate regret with approximation factor $c_\alpha = (1-\alpha)^{-(1-\alpha)} \leq 1.445$, for $0 \leq \alpha < 1$. The upper bound on the $c_\alpha$-regret for this problem exhibits a surprising *phase transition* phenomenon – transitioning from a power-law to a constant at the critical exponent $\alpha = 1/2$. Our result also resolves an open problem in designing an efficient no-regret policy for the online job scheduling problem in certain parameter regimes. Along the way, we introduce new algorithmic and analytical techniques, including greedy estimation of the future gradients for non-additive global reward functions and bootstrapping second-order regret bounds, which may be of independent interest.

## 1 Introduction

The notion of *algorithmic fairness* refers to learning algorithms that guarantee fair predictions even when subjected to adversarially biased training data [Dwork et al., 2012]. Fairness has become a major criterion for designing and deploying large-scale learning algorithms that affect a diverse user base. Since the training data could be highly skewed in practice, it is essential to make minimal assumptions about the data-generating process and design provably robust and fair learning policies. Guaranteeing fairness becomes even more challenging in the online learning setup as there is no distinction between the training and test data, and no assumption is made on the input data sequence. As an example, consider an online recruitment campaign where a learning algorithm decides the target group (identified by, say, the tuple (`race`, `gender`, `age`)) to which an ad for a job vacancy is to be displayed. Suppose a revenue-maximizing recommendation algorithm concludes from past data that more revenue is generated by showing the ad to Group A compared to Group B. In that case, the ad-serving algorithm will eventually end up showing that ad exclusively to Group A while discriminating against Group B users of a potential job opportunity [Hao, 2019]. One of the overarching goals of this paper is to design efficient online learning policies that provably mitigate this algorithmic bias while maintaining efficiency *irrespective* of the past data seen so far (refer to Example B.1 in the Appendix for a closely-related scheduling problem considered in this paper).

37th Conference on Neural Information Processing Systems (NeurIPS 2023).

Towards this goal, we consider a generic online fair resource allocation problem, which we call NOFRA (No-Regret Fair Resource Allocation). In this problem, a fixed set of resources needs to be equitably shared among $m$ agents over multiple rounds. Note that fairness is a complex, multidimensional, and essentially subjective concept. Several metrics have been introduced in the literature to quantify the degree of fairness in resource allocation, including $\alpha$-fairness [Lan et al., 2010], proportional fairness [Kelly, 1997, Mo and Walrand, 2000], max-min fairness [Radunovic and Le Boudec, 2007, Nace and Pióro, 2008], and Jain's fairness index [Jain et al., 1984]. In this paper, we consider the problem of maximizing the $\alpha$-fairness function, in which the utility of each agent when allocated $R$ units grows as $R^{1-\alpha}$. The parameter $\alpha$ is restricted to the interval $[0, 1)$. This same range of $\alpha$ has been studied earlier in a game-theoretic set up by Altman et al. [2010].

**On the hardness of the NOFRA problem:**   In the standard online learning setting, the cumulative reward accrued over a given time horizon is the sum of the rewards obtained at each round [Hazan, 2019]. In contrast, the objective in NOFRA is to maximize the global $\alpha$-fairness function, which is equal to the sum of cumulative rewards of each agent raised to the power $1 - \alpha, 0 \leq \alpha < 1$. Due to the power-law non-linearity, the objective function of NOFRA is non-separable with respect to time. Note that the non-separability of the objective function is essential to induce fairness in sequential allocations by incorporating the diminishing return property. However, this renders the NOFRA problem fundamentally different from standard online learning problems. In particular, Theorem 2 proves a non-trivial lower bound to the approximation ratio achievable by any online learning policy for this problem. This lower bound does not hold in classic settings with additive rewards.

**Our contributions:**   We show that despite its non-separable structure, the NOFRA problem can be approximately reduced to an online linear optimization problem with a greedily defined sequence of reward vectors. The resulting problem can then be solved with an online convex optimization (OCO) policy with a second-order regret bound. In particular, we make the following contributions:

- In Algorithm 1, we present an efficient online resource allocation policy, Online Fair Allocation (OFA), that approximately maximizes the aggregate $\alpha$-fairness function of the agents. We show that OFA achieves $(1 - \alpha)^{-(1-\alpha)} \leq 1.445$-approximate sublinear regret (Theorem 1). To the best of our knowledge, OFA is the first online policy that approximately maximizes the $\alpha$-fairness against *any* adversary, which could even be adaptive.

- In Theorem 2, we establish a lower bound to the approximation factor $c_\alpha$ achievable by any online learning policy for the $\alpha$-fair reward function. Our lower bound improves upon the prior best-known lower bound of Si Salem et al. [2022], which established only that $c_\alpha > 1$.

- Technically, our proposed OFA policy optimizes the non-additive global reward function by greedily estimating the future gradients and using these estimates to construct an online linear optimization problem such that sublinear regret for this linear problem translates into sublinear approximate-regret for the global optimization problem. The linear optimization problem is then solved with an online gradient ascent policy with adaptive step sizes. The resulting algorithm is simple and intuitive: we have weights on each user that decay as the user's cumulative reward increases.

- On the analytical side, a key challenge is that the online linear optimization problem has a demand sequence (and hence, a gradient sequence) that depends on our past actions. We introduce a new proof technique that controls the norm of the gradients in this setting so that a tight second-order regret bound can be established. This technique applies generally to problems with states where the future gradients depend on past actions.

- We provide numerical simulation results supporting our theoretical conclusions and demonstrating the effectiveness of our approach in several applications of the NOFRA problem.

**Related work:**   Several closely related works with non-additive reward functions appear in the online learning literature [Si Salem et al., 2022, Even-Dar et al., 2009, Rakhlin et al., 2011]. In the following, we explain why the NOFRA problem is fundamentally different from the existing studies. An extended literature review is given in Appendix A.

Closest to our work, Si Salem et al. [2022] considered the problem of designing fair online resource allocation policies. The authors showed that it is impossible to design a no-regret policy without restricting the set of admissible adversarial demand sequences. Given this negative result, the authors

proposed a no-regret policy using a primal-dual framework under the assumption of a restricted adversary, which exhibits an essentially i.i.d. stochastic-like fluctuation. In contrast, we design a robust policy with an approximate sublinear regret *without restricting* the adversary. Due to the significantly weaker assumption, our proposed policy and its analysis are drastically different from that of Si Salem et al. [2022]. Even-Dar et al. [2009] considered maximizing a global concave objective function in the no-regret setup under the further assumption that the optimal offline reward is convex. They presented an approachability-based policy in this setting and also proved the impossibility of achieving sublinear regret when the convexity condition is violated. Although the objective function in NOFRA is concave, we show that the optimal static offline reward function fails to be convex (see Appendix C.8). Hence, the policy of Even-Dar et al. [2009] does not apply to the NOFRA problem. Rakhlin et al. [2011] considered the problem of no-regret learnability for a wide class of non-additive global functions. However, their results also do not apply to our setting as our problem is not no-regret learnable (see Theorem 2). Agrawal et al. [2016] considered the problem of maximizing a concave utility function with the bandit information feedback and derived an efficient policy. However, their $\tilde{O}(\sqrt{T})$ regret bound holds only in the stochastic setting. Offline fair caching algorithms using the $\alpha$−fair utility function were proposed in Liu et al. [2020]. However, their policy cannot be used in the online setting due to the casually available information structure. Finally, the work by Paria and Sinha [2021], Bhattacharjee et al. [2020], Mukhopadhyay et al. [2022] considered the online caching problem with the linear utility function. However, due to the non-linearity, extending their work to the $\alpha$-fair utility function entails several technical challenges, which we resolve in this paper.

## 2 Problem Formulation

In this section, we give the general formulation of the NOFRA problem, along with examples of concrete resource allocation problems that fit into this framework.

Consider a set of *agents* among which a limited resource is to be fairly divided. Assume that the resource allocated to the $i^{\text{th}}$ agent on the $t^{\text{th}}$ round is represented by an $N$-dimensional non-negative vector $y_i(t)$, where $N$ is arbitrary. On every round $t$, the $i^{\text{th}}$ agent requests an $N$-dimensional non-negative *demand* (or, reward) vector $x_i(t)$. The demand vectors are revealed to an online allocation policy $\pi$ at the end of each round. We make *no assumption* on the regularity of the demand vector sequence, which could be adversarial (*c.f.,* Si Salem et al. [2022]). Before the $N \times m$ dimensional aggregate demand matrix $\boldsymbol{x}(t) \equiv \big(x_1(t), x_2(t), \ldots, x_m(t)\big)$ for round $t$ is revealed, the online resource allocation policy $\pi$ chooses a non-negative $N \times m$ dimensional allocation matrix $\boldsymbol{y}(t) = \big(y_1(t), y_2(t), \ldots, y_m(t)\big)$ from the set of all feasible allocations $\Delta$. The set of all feasible allocations is assumed to be convex (see Remark 2 below). The reward accrued by the agent $i$ on round $t$ is given by the inner-product $\langle x_i(t), y_i(t) \rangle$, which, without loss of generality, is assumed to be upper-bounded by one. The cumulative reward accrued by agent $i$ by the end of round $t$ is given by

$$R_i(t) = R_i(t-1) + \langle x_i(t), y_i(t) \rangle, \ \forall i, t, \tag{1}$$

where we set the initial condition $R_i(0) \equiv 1, \forall i.$[1] By iterating the above recursion, the cumulative reward $R_i(t)$ can be alternatively expressed as follows:

$$R_i(t) = 1 + \sum_{\tau=1}^{t} \langle x_i(\tau), y_i(\tau) \rangle. \tag{2}$$

We make two mild technical assumptions on the structure of the demand and allocation vectors.

**Assumption 1.** $\delta \leq \|x_i(t)\|_1 \leq 1, \forall i, t,$ *for some constant* $\delta > 0$.

**Assumption 2.** *Let* $\mathbf{1}_{N \times m}$ *denote the* $N \times m$ *all-1 matrix. Then* $\mu \mathbf{1}_{N \times m} \in \Delta$ *for some constant* $\mu > 0$.

The above assumptions imply that it is possible to ensure a non-zero reward for all agents on all rounds. The assumptions are used in the regret analysis only (see Eq. (25)). Our proposed online policy is oblivious to the value of the parameters $\delta$ and $\mu$.

The utility of any user for a cumulative reward of $R$ is given by the concave $\alpha$-fair utility function $\phi : \mathbb{R}_+ \rightarrow \mathbb{R}_+$, defined as follows:

$$\phi(R) = \phi_\alpha(R) \equiv \frac{R^{1-\alpha}}{1-\alpha}, \ \ R \geq 0, \tag{3}$$

---

[1]The rationale behind setting the initial cumulative rewards to non-zero values is to ensure that the utility function $\phi$ remains continuously differentiable at any round $t \geq 1$.

for some constant $0 \leq \alpha < 1$.[2] The fairness parameter $\alpha$ induces a trade-off between the desired efficiency and fairness by incorporating a notion of *diminishing return* property in the global objective function. The static offline optimal allocation with larger $\alpha$ leads to more equitable cumulative rewards [Bertsimas et al., 2012]. Setting $\alpha = 0$ reduces the problem to the "unfair" online linear optimization problem. Our objective is to design an online resource allocation policy $\pi \equiv \{\boldsymbol{y}(t)\}_{t \geq 1}$ that minimizes the regret for maximizing the aggregate utilities of all users compared to any fixed offline resource allocation strategy $\boldsymbol{y}^* \in \Delta$. To be precise, assume that the offline, fixed resource allocation $\boldsymbol{y}^*$ yields a cumulative reward of $\boldsymbol{R}^*(T)$ for a horizon of length $T \geq 1$. Then, our objective is to design a resource allocation policy which minimizes the $c$-approximate regret (which we refer to as $c$-regret for short) defined as:

$$\text{Regret}_T(c) \equiv \sum_{i=1}^{m} \phi(R_i^*(T)) - c \sum_{i=1}^{m} \phi(R_i(T)),\tag{4}$$

for some small constant $c \geq 1$.[3] In the case of standard regret ($c = 1$), we drop the argument in the parenthesis. Note that, unlike the standard online convex optimization problem, in the NOFRA problem, the reward function is global, in the sense that it is non-separable across time [Even-Dar et al., 2009]. For this reason, it is necessary to consider the $c$-regret with $c > 1$, rather than the standard regret (i.e., with $c = 1$). This will become clear from Theorem 2, where we prove an explicit lower bound to the approximation factor achievable by any online policy for the global $\alpha$-fair reward function. This lower bound implies a concrete lower bound on the achievable $c$. Our lower bound improves upon [Si Salem et al., 2022, Theorem 1], where it was shown that no online policy can achieve a sublinear standard regret for the NOFRA problem under an unrestricted adversary.

**Remark 1:** When $\alpha > 1$, the offline benchmark $\phi(R_i^*(T))$ itself becomes $O(1), \forall i$. Hence, in this regime, a sublinear regret bound (4) becomes vacuous. Consequently, we restrict the fairness parameter $\alpha$ to the interval $[0, 1)$.

**Remark 2:** In the above problem definition, we assumed that the set of feasible allocations $\Delta$ is convex and thus that there are no integrality constraints on the allocation, *i.e.,* the components of the allocation matrix $\boldsymbol{y}(t)$ are allowed to be fractional. However, in many combinatorial resource allocation problems, the allocation vector is required to be integral. In this case, the feasible action set $\Delta$ is naturally defined to be the convex hull of the integral actions. In Appendix C.6, we consider the setting with integrality constraints and extend our algorithm and analysis to design a randomized integral allocation policy with a sublinear regret bound.

**Motivating examples:** The statement of NOFRA is fairly general, and by suitably choosing the reward and allocation vectors, many standard resource allocation problems can be reduced to NOFRA. In what follows, we highlight the example of fair shared caching [Bhattacharjee et al., 2020, Paria and Sinha, 2021]. We refer the readers to Appendix B for additional examples of online job scheduling and online matching.

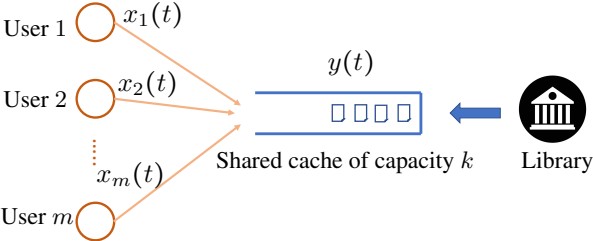

Figure 1: The online shared caching problem

[2] Some authors define the $\alpha$-fairness function with an extra additive term, *i.e.,* $\phi_\alpha'(R) = \frac{R^{1-\alpha}-1}{1-\alpha}$ [Si Salem et al., 2022]. Note that, in the range $0 \leq \alpha < 1$, this alternative definition changes the regret metric (4) only by an additive constant. We use the fairness function $\phi_\alpha(R)$ as it is *positively homogeneous* of degree $(1 - \alpha)$ - a property that we exploit in our analysis.

[3] Our theoretical results and algorithms trivially extend to the $\boldsymbol{w}$-weighted fairness function $\sum_i w_i \phi(R_i(T))$ for some non-negative weight vector $w_i \geq 0, \forall i$.

In the *Online Shared Caching* problem with a single shared cache, $m$ users are connected to a single cache of capacity $k$ (see Figure 1 for a schematic). At each round, each user requests a file from a library of size $N$. The file request sequence may be adversarial. At the beginning of each round $t$, an online caching policy prefetches at most $k$ files on the cache, denoted by the vector $y(t)$ such that

$$\sum_{i=1}^{N} y_i(t) = k, \ \ 0 \le y_i(t) \le 1, \forall i \in [N]. \tag{5}$$

The set of all feasible caching configurations is denoted by $\Delta_k^N$.[4] Immediately after the prefetching at round $t$, each user $i$ reveals its file request, represented by the one-hot encoded demand vector $x_i(t)$, $1 \le i \le m$. A special case of the above caching model for a single user ($m = 1$) has been investigated in previous work [Bhattacharjee et al., 2020, Mhaisen et al., 2022, Joshi and Sinha, 2022].

Let $R_i(t)$ denote the cumulative (fractional) hits obtained by the $i^{\text{th}}$ user up to round $t \ge 1$. We have

$$R_i(T) = 1 + \sum_{t=1}^{T} \langle x_i(t), y(t) \rangle, \ \ 1 \le i \le m. \tag{6}$$

The online shared caching problem can be easily reduced to an instance of the NOFRA problem by taking the demand matrix to be $\boldsymbol{x}(t) = \big(x_1(t), x_2(t), \dots, x_m(t)\big)$. Since the allocation vector $y(t)$ is common to all users, the allocation matrix can be taken to be $\boldsymbol{y}(t) = \big(y(t), y(t), \dots, y(t)\big)$. It can be observed that Assumption 2 holds in this case with $\mu = k/N$ by noting that $k/N \mathbf{1}_N \in \Delta_k^N$.

## 3 Designing an approximately-no-regret policy for NOFRA

In view of the difficulty outlined in Section 1, the design and analysis of the Online Fair Allocation (OFA) policy consists of two parts. First, in Lemma 1, we show that by greedily estimating the terminal gradient with the current gradients at each round, the $(1-\alpha)^{-(1-\alpha)}$-approximate regret of the NOFRA problem can be upper bounded by the standard regret of a surrogate online linear optimization problem with a policy-dependent gradient sequence. Note that the regret of the surrogate problem depends on the norms of the gradients, which, in turn, depends on the policy. We use the online gradient ascent policy with adaptive step sizes to solve the surrogate problem while simultaneously controlling the norm of the gradients. The following section details our technique.

### 3.1 Reducing the NOFRA problem to an Online Linear Optimization Problem with policy-dependent subgradients

Since the $\alpha$-fairness function $\phi : \mathbb{R}_{>0} \to \mathbb{R}_{>0}$ is concave and continuously differentiable, we have that for any $x, y > 0$:

$$\phi(x) - \phi(y) \le \phi'(y)(x - y). \tag{7}$$

Let $\beta \ge 1$ be a constant, which will be fixed later. Taking $x = R_i^*(T)$ and $y = \beta R_i(T)$ in the above inequality, we have

$$
\begin{aligned}
\phi(R_i^*(T)) - \beta^{1-\alpha} \phi(R_i(T)) \ &\stackrel{(a)}{=} \ \phi(R_i^*(T)) - \phi(\beta R_i(T)), \\
&\stackrel{(b)}{\le} \ \phi'(\beta R_i(T))(R_i^*(T) - \beta R_i(T)) \\
&\stackrel{(c)}{\le} \ \beta^{-\alpha} \phi'(R_i(T)) \sum_{t=1}^{T} \langle x_i(t), y_i^* - \beta y_i(t) \rangle, \tag{8}
\end{aligned}
$$

where in (a), we have used the positive homogeneity property of the fairness function $\phi(\beta x) = \beta^{1-\alpha} \phi(x)$, in (b), we have used the concavity of the fairness function $\phi(\cdot)$ from Eqn. (7), and, finally, in (c), we have used the definition of cumulative rewards (2), the fact that $\beta \ge 1$, and the homogeneity of the function $\phi(\cdot)$, which gives $\phi'(\beta x) = \beta^{-\alpha} \phi'(x)$.

---

[4]In the above, we allow fractional caching, which can be easily converted to a randomized integral caching policy (where $y_i(t) \in \{0, 1\}, \forall i, t$) via sampling. See Section C.6 for details.

Summing up the bound (8) over all agents $i \in [m]$, we obtain the following upper bound to the $\beta^{1-\alpha}$-Regret of any online policy for the NOFRA problem:

$$\text{Regret}_T(\beta^{1-\alpha}) \le \beta^{-\alpha} \sum_{t=1}^{T} \left( \sum_i \langle \phi'(R_i(T))x_i(t), y_i^* - \beta y_i(t) \rangle \right). \tag{9}$$

It is critical to note that each of the quantities $\phi'(R_i(T))$ depends on the entire sequence of past actions $\{y_t\}_{t=1}^{T}$. Hence, $\phi'(R_i(T))$ in (9) is not known to the online policy when it takes its actions. More importantly, the value of the coefficient $\phi'(R_i(T))$ depends on the future actions and requests through (1). Hence, it is *impossible* to know the value of this coefficient in an online fashion. To get around this fundamental difficulty, we now consider a *surrogate* regret-minimization problem by replacing the term $\phi'(R_i(T))$ by its *causal counterpart* $\phi'(R_i(t-1))$ at the beginning of round $t$. In other words, we now design an online learning policy that minimizes the regret of the following online linear optimization problem:

$$\text{L-Regret}_T \equiv \sum_{t=1}^{T} \left( \sum_i \langle \phi'(R_i(t-1))x_i(t), y_i^* - y_i(t) \rangle \right). \tag{10}$$

To recapitulate the information structure, recall that at the beginning of round $t$, the cumulative reward vector $\boldsymbol{R}(t-1)$, and hence the coefficient $\phi'(R_i(t-1))$, is known. Still, the current request vector $\boldsymbol{x}(t)$ is unknown to the online learner before it makes its allocation decision $y_t$ on the $t^{\text{th}}$ round. We now establish the following key lemma that relates the original regret bound (9) to the regret bound (10) for the surrogate problem.

**Lemma 1.** *Consider any arbitrary online resource allocation policy and assume that the utility function of each user is given by the $\alpha$-fair utility function (3). Then, for any horizon of length $T \ge 1$, we have*

$$\text{Regret}_T(c_\alpha) \le (1-\alpha)^\alpha \cdot \text{L-Regret}_T + c_\alpha m, \tag{11}$$

*where $c_\alpha \equiv (1-\alpha)^{-(1-\alpha)} \le e^{1/e} < 1.445$.*

**Proof outline:** As discussed above, the surrogate problem greedily replaces the non-causal terminal gradient $\phi'(R_i(T))$ in Eqn. (9) by the current gradient $\phi'(R_i(t-1)), \forall i$ on round $t$. To prove Lemma 1, we show that this transformation can be done by incurring only a small penalty factor to the overall regret bound. To show this, we split the regret expression (9) into the difference between two terms - term $(A)$ corresponding to the cumulative reward accrued by the static allocation $(\boldsymbol{y}^*)$, and term $(B)$ corresponding to the reward accrued by the online policy. Next, we compare these two terms separately with the corresponding terms $(A')$ and $(B')$ in the regret expression for the surrogate online linear optimization problem (10). By exploiting the non-decreasing nature of the cumulative rewards, we first show that $(A) \le (A')$ (Eq. (15)). Next, by using properties of the $\alpha$-fair utility function, we show that $(B') \le (1-\alpha)^{-1}(B + m)$ for any policy (Eq. (16)). Lemma 1 then follows by combining these two results. See Appendix C.2 for the proof.

## 3.2 Online Learning Policy for the Surrogate Problem and its Regret Analysis

Having established that the $c_\alpha$-regret of the original problem is upper-bounded by the regret of the surrogate learning problem, we now proceed to upper bound the regret of the surrogate problem under the action of a no-regret learner with a second-order regret bound. In the sequel, we use the projected Online Gradient Ascent (OGA) with adaptive step sizes [Orabona, 2020, Algorithm 2.2] for designing a no-regret policy for the surrogate problem. We call the resulting online learning policy ONLINE FAIR ALLOCATION (OFA). The pseudocode of the OFA policy is given in Algorithm 1.

**The Online Fair Allocation policy (OFA):** Since $\phi'(R_i(t-1)) = 1/R_i(t-1)^\alpha$, at the end of round $t$, each user $i$ computes its gradient component $g_i$ by dividing its current demand vector $x_i(t)$ with its current cumulative reward $R_i(t-1)$ raised to the power $\alpha$ (line 5 of Algorithm 1). This formalizes the intuition that a user with a larger current cumulative reward has a smaller gradient component. In line 8, we take a projected gradient ascent step. Depending on the problem, the projection can often be efficiently computed, *e.g.,* using variants of the Frank-Wolfe algorithm given access to an efficient LP

---

[5]See Appendix C.1 for upper bounds on the Euclidean diameter of the action sets for Examples 2.1-2.3.

---

**Algorithm 1** The Online Fair Allocation (`OFA`) Policy

---

1: **Input:** Fairness parameter $0 \le \alpha < 1$, Demand/Reward vectors from the agents $\{\boldsymbol{x}(t)\}_{t=1}^T$, Euclidean projection oracle $\Pi_\Delta(\cdot)$ onto the feasible set $\Delta$, an upper-bound $D$ to the Euclidean diameter of the feasible set[5].
2: **Output:** Online resource allocation decisions $\{y_t\}_{t=1}^T$
3: $R_i \leftarrow 1, \forall i \in [m], S \leftarrow 0$             ▷ *Initialization*
4: **for each** round $t = 2 : T$: **do**
5:    $g_i \leftarrow \frac{x_i(t-1)}{R_i^\alpha}, \forall i \in [m]$      ▷ *Computing the gradient components for each agents*
6:    $g \leftarrow (g_1, g_2, \dots, g_m)$          ▷ *Computing the full gradient*
7:    $S \leftarrow S + \|g\|_2^2,$          ▷ *Accumulating the norm of the gradients*
8:    $y \leftarrow \Pi_\Delta\left(y + \frac{D}{2\sqrt{S}}g\right)$      ▷ *Updating the inclusion probabilities using OGA*
9:    [**Optional**] Sample a randomized integral allocation $Y$ s.t. $\mathbb{E}[Y] = y$.
10:    The agents reveal their demand/ reward vectors $\{x_i(t)\}_{i \in [m]}$ for the current round.
11:    $R_i \leftarrow R_i + \langle x_i(t), y_i \rangle, \ \forall i \in [m].$     ▷ *Updating cumulative rewards*
12: **end for each**

---

oracle. With additional structure, `OFA` can be simplified further with an even more efficient projection. For example, we give a simplified implementation for the online shared caching problem in Appendix C.5 by exploiting the fact that the action vector $y(t)$ is the same for all users. Step 9 is an optional sampling step which is executed only if an integral allocation is required. We discuss this step in Appendix C.6. Finally, the cumulative rewards of all users are updated in Step 11. The following lemma gives an upper bound to the regret of the `OFA` policy for the surrogate problem.

**Lemma 2.** *The Online Fair Allocation policy, described in Algorithm 1 achieves the following standard regret bound for the surrogate problem* (10) *for the $\alpha$-fair utility function:*

$$
\textit{L-Regret}_T = \begin{cases} O(T^{1/2-\alpha}), & \textit{if } 0 < \alpha < 1/2, \\ O(\sqrt{\log T}), & \textit{if } \alpha = 1/2 \\ O(1), & \textit{if } 1/2 < \alpha < 1. \end{cases}
$$

*Further, under this policy, the cumulative rewards of each user increase linearly with time,* i.e., $R_i(T) = \Omega(T), \forall i, T.$

**Proof outline:** One of the major challenges in the regret analysis of the surrogate problem (10) is that the coefficients of the gradients $\{\phi'(R_i(t-1)), i \in [m]\}$ on round $t$ depends on the past actions $\{\boldsymbol{y}(\tau)\}_{\tau=1}^t$ of the policy itself. Since the second-order regret bound of any online linear optimization problem scales with the norm of the gradients, we now need to *simultaneously* control the regret *and* the norm of the gradients generated by the online policy. Surprisingly, the proof of Lemma 2 shows that the proposed `OGA` policy with adaptive step sizes not only provides a sublinear regret but also keeps the gradients small, which in turn helps keep the regret small. In fact, these two goals are well-aligned to each other, and our proof exploits the reinforcing nature of these two objectives via a new *bootstrapping* technique. See Appendix C.4 for the proof.

Finally, combining Lemma 1 and Lemma 2, we obtain the main result of this paper.

**Theorem 1.** *The Online Fair Allocation (`OFA`) policy, described in Algorithm 1, achieves the following approximate regret bound for the `NOFRA` problem* (4) *with $c_\alpha \equiv (1-\alpha)^{-(1-\alpha)}$:*

$$
\textit{Regret}_T(c_\alpha) = (1-\alpha)^\alpha \begin{cases} O(T^{1/2-\alpha}) & \textit{if } 0 < \alpha < 1/2, \\ O(\sqrt{\log T}) & \textit{if } \alpha = 1/2 \\ O(1) & \textit{if } 1/2 < \alpha < 1, \end{cases} \tag{12}
$$

**Remarks:** 1. For the job scheduling problem in the reward maximization setting, Even-Dar et al. [2009, Lemma 4] showed that if the cumulative reward is concave and the offline optimal reward is convex, then their proposed approachability-based recursive policy, whose complexity scales exponentially with the number of machines $m$, achieves sublinear regret. In Section 7 of the same paper, the authors posed an open problem of attaining a relaxed goal when the above sufficient condition is violated. In Appendix C.8, we show that for the $\alpha$-fair utility function (3),

the offline optimal cumulative reward is *non-convex* in the regime $0 < \alpha < 1$. Hence, Theorem 1 gives a resolution to the above open problem by exhibiting a simple online policy with a sublinear approximate regret when the given sufficient condition is violated. Furthermore, the computational complexity of our policy is linear in the number of machines $m$, which is significantly lower than that of their approachability-based policy, whose complexity scales exponentially fast in $m$.

2. Observe that the regret bound given by Theorem 1 always remains non-vacuous, *irrespective* of the value of $\alpha$ and the sequence of adversarial reward vectors. This follows from the fact that irrespective of the demand vectors, by choosing the constant action $y = \mu \mathbf{1}_{N \times m}$ (which is feasibly by Assumption 2), each user can achieve a cumulative reward of $\mu \delta T$. Hence, the optimal offline value of the $\alpha$-fair utility function is $\Omega(T^{1-\alpha})$. On the other hand, by Theorem 1, the OFA policy achieves a regret bound of $O(T^{1/2-\alpha})$, which is always dominated by the optimal static offline objective.

3. When $\alpha = 0$, NOFRA corresponds to the cumulative reward maximization problem for all users. Hence, Theorem 1 recovers the well-known $O(\sqrt{T})$ standard regret bound [Orabona, 2020].

The following converse result gives a universal lower bound to the approximation factor $c$ for which it is possible to design an online policy for NOFRA with a sublinear $c$-regret.

**Theorem 2** (Lower bound on approximation factor). *Consider the online shared caching problem for the $\alpha$-fair reward function with $m = 2$ users. Any online policy with a sublinear $c_\alpha$-regret must have*

$$c_\alpha \geq \max_{0 \leq \eta \leq 1/2} \frac{\eta^{1-\alpha} + (1-\eta)^{1-\alpha}}{(1-\eta/2)^{1-\alpha} + (\eta/2)^{1-\alpha}} > 1, \ \ 0 < \alpha < 1.$$

See Appendix C.9 for the proof of Theorem 2. A numerical comparison between the upper and lower bounds on the approximation factor is shown in Figure 7 in the Appendix. Observe that for $\alpha \in (0, 1)$ the above lower bound is strictly greater than 1, improving on the prior best known lower bound of [Si Salem et al., 2022], which showed just that sublinear standard regret (i.e., $c_\alpha = 1$) is unachievable.

## 4   Experimental results

In this section, we report experimental results using the fair caching problem as a case study[6]. Additional experiments on fair scheduling are provided in Appendix D.2. We compare the performance of our algorithms on two datasets against several baselines showing the effectiveness of our algorithm.

**Setup and Dataset:**   We perform simulations on both synthetically generated data and on CDN traces from Berger [2018]. For our synthetic dataset, the request patterns of the users are highly homogeneous, letting us demonstrate the fairness behavior of different algorithms. In particular, we take time horizon $T = 1000$ rounds, number of users $m = 5$, cache size $C = 7$, and library size $N = 30$. At each time step: (1) users 1 and 2 request a file from file numbers 0-29 uniformly at random; (2) user 3 repeatedly requests file numbers 0-3; (3) user 4 repeatedly requests file numbers 4-18; (4) user 5 repeatedly requests file numbers 19-27. Intuitively, the difficulty comes in ensuring a fair allocation of hits to users 1 and 2, who request files from a larger space of possibilities than the other users.

The CDN dataset Berger [2018] contains a request ID for each request, corresponding to the timestamp and the file ID of the requested file. We set $T = 400$ rounds, number of users $m = 4$, cache size $C = 10$ and, library size $N = 50$. For preprocessing, first, we sort the data according to the timestamp and discard all the files with an ID greater than $N$, and then only consider the first $m \times T = 1600$ requests among the remaining requests. We then allocate files to users in decreasing order of their popularity, such that user 1 tends to request common files, while user 4 tends to request less common files. This makes ensuring a fair cache allocation difficult. See Appendix D.1 for details.

**Comparison:**   We compare our algorithm with the *Online Horizon-Fair* (OHF) policy proposed by Si Salem et al. [2022]. Just like the OFA policy, OHF is also an online policy that ensures long-term fairness using the $\alpha$-fairness utility, which makes it an ideal candidate for comparison. We also compare our algorithm with the commonly used Least-Recently-Used (LRU) and Least-Frequently-Used (LFU) cache replacement policies. We compare our algorithm with the fixed offline optimal allocation and the maximin optimal allocation. Both policies have access to all the file requests ahead

---

[6]The code is available at: `https://github.com/AtivJoshi/nofra`

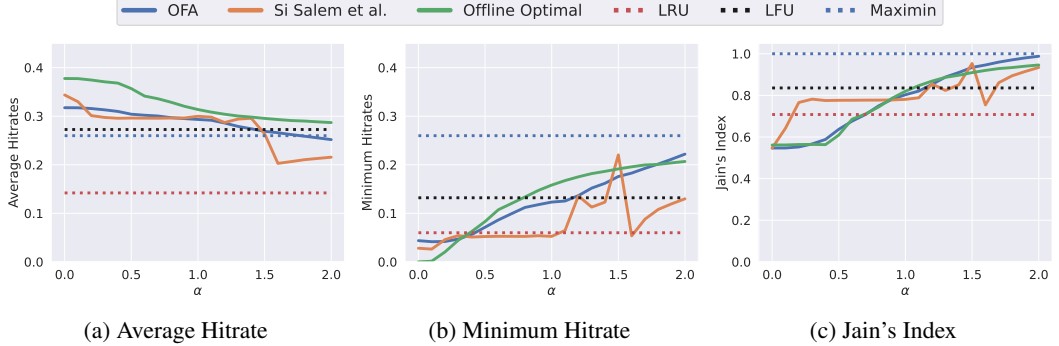

(a) Average Hitrate       (b) Minimum Hitrate       (c) Jain's Index

Figure 2: Synthetic Dataset

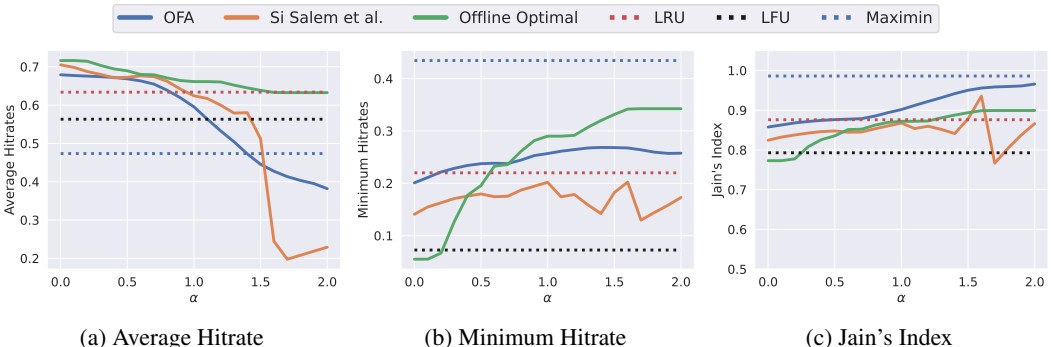

(a) Average Hitrate       (b) Minimum Hitrate       (c) Jain's Index

Figure 3: CDN Dataset

of time. The fixed optimal offline allocation is given by $y^* = \operatorname{argmax}_{y \in \mathcal{Y}} \sum_{i=1}^{m} \phi(R_i(T))$, where $\mathcal{Y}$ is the set of feasible cache configurations (see Eqn. (5)) and $R_i(T)$ is defined in Eqn. (6). The maximin optimal allocation is the fixed offline caching configuration that maximizes the minimum hitrate among all users. It can be computed via an LP. We compare these algorithms in terms of the average hit rate, minimum hit rate, and Jain's fairness index as the value of $\alpha$ changes. Jain's fairness index is a common metric used to quantify if the users are receiving a fair share of network resources. In our case, the fairness index is given by

$$\mathcal{J}(R_1(T), \ldots, R_m(T)) = \frac{(\sum_{i=1}^{m} R_i(T))^2}{m \sum_{i=1}^{m} R_i^2(T)}.$$

The index ranges from $1/m$ for minimum fairness to 1 for maximum fairness.

**Observations:** The average hitrates of each policy are shown in Figures 2a and 3a. For the synthetic dataset, despite being fair, the overall performance of the OFA policy is at par with the traditional caching algorithms like LRU and LFU policies. For the CDN dataset, the OFA policy initially performs close to the Si Salem et al. [2022] algorithm while for $\alpha \geq 1.5$, the OFA thoroughly outperforms it. For $\alpha > 1$, there is a drop in performance of OFA as compared to the "unfair" LRU and LFU algorithms.[7] The minimum hitrates of each policy are shown in Figures 2b and 3b. For both datasets, the OFA policy outperforms Si Salem et al. [2022], LRU, and LFU. We can also see the minimum and average hitrates for the OFA policy come closer to each other as we increase $\alpha$. I.e. the algorithm becomes fairer at the cost of total hitrate. Finally, from the plots in Figures 2c and 3c, we can see that as $\alpha$ increases, OFA outperforms all other algorithms in terms of Jain's fairness index. Also, Jain's index for OFA approaches the maximum value of 1 as we increase $\alpha$.

---

[7]Although regret guarantees for $\alpha > 1$ become vacuous, to clearly demonstrate the fairness characteristics of algorithms, we run simulations for $\alpha \in [0, 2]$.

# 5 Conclusion and open problems

In this paper, we propose an efficient online resource allocation policy, Online Fair Allocation (OFA), that achieves a $c_\alpha$-approximate sublinear regret bound for the $\alpha$-fairness objective, where $c_\alpha \equiv (1-\alpha)^{-(1-\alpha)} \leq 1.445$, for $0 < \alpha < 1$. Our main technical contribution is to show that the non-additive $\alpha$-fairness function can be efficiently learned by greedily estimating the terminal gradients. An important follow-up problem is to investigate the extent to which the algorithmic and analytical methodologies introduced in this paper can be generalized. Note that, for the online scheduling problem, only a recursive approachability-based policy is known in the literature, whose complexity scales *exponentially* with the number of machines [Even-Dar et al., 2009]. The algorithm and analysis presented in this paper are specific to the $\alpha$-fair utility function. It would be interesting to investigate whether these ideas can be extended to learning general concave utility functions. Another related problem is to design an optimistic version of the proposed OFA policy that offers an improved regret bound by efficiently incorporating hints regarding the future demand sequence [Mhaisen et al., 2022, Bhaskara et al., 2020]. Finally, reducing the gap between the upper and lower bounds of the approximation factor in Figure 7 would be of interest.

## 6 Acknowledgement

This work is supported by a US-India NSF-DST collaborative grant coordinated by IDEAS-Technology Innovation Hub (TIH) at the Indian Statistical Institute, Kolkata, and as a supplementary fund of NSF CAREER 2045641. A. Sinha was additionally supported by the Qualcomm Research Grant IND-417880. M. Hajiesmaili's work is supported by CNS-2102963, CNS-2106299, and CPS-2136199. C. Musco was also partially supported by an Adobe Research grant.

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

# No-regret Algorithms for Fair Resource Allocation

**A. Sinha, A. Joshi, R. Bhattacharjee, C. Musco, M. Hajiesmaili**

## A  Related Work

We provide a more comprehensive review of the fair machine learning literature in this section. Multiple different definitions have been used to quantify the fairness of machine learning algorithms. Hardt et al. [2016] introduced equality of opportunity as a fairness criterion, which ensures that individuals have an equal chance of being correctly classified by machine learning algorithms, regardless of their protected attributes like race or gender. Kleinberg et al. [2017] formalized three different notions of fairness and showed that no algorithm can satisfy these notions simultaneously, thus showing the inherent trade-offs in competing notions of fairness. Other prevalent fairness criteria include Price-of-fairness introduced by Bertsimas et al. [2011] which quantifies how much the aggregate utility is affected by enforcing fairness. Corbett-Davies and Goel [2016] discussed the limitations of different notions of fairness widely used in the fair machine learning literature. There have been different works that examined the societal harm caused by unfair algorithms in critical domains. For example, Chouldechova [2017] investigated the disparate impact of recidivism prediction algorithms used in the criminal justice system on different demographic groups. Since our work on online fair resource allocation is closely related to fairness in online algorithms, job scheduling, and matching, we provide a more detailed review of papers related to fair algorithms in each of these domains below.

**Fairness in online algorithms:** Several works have considered fairness specifically in the online setting using both regret and competitive ratio as performance metrics. Sinclair et al. [2022] considered the problem of allocating resources to individuals in an online setting, where the number and type of individuals arriving in each round are drawn from some fixed known distribution. The allocations must balance envy-freeness (requiring that each agent prefers the agent's own allocation over the other's allocation) and efficiency (the amount of consumed resources' budget). Banerjee et al. [2022] design competitive algorithms for the problem of allocating a set of divisible goods to agents in an online manner while maximizing the Nash social welfare, which is another metric that quantifies the trade-off between fairness and efficiency. They also study the setting when the algorithm is allowed to access some predictions related to the utility function of each agent over the goods. Arsenis and Kleinberg [2022] introduce notions of individual fairness in optimal stopping problems and provide competitive algorithms that satisfy these definitions of fairness.

There have been different works that consider fairness in terms of regret. Talebi and Proutiere [2018] studied a setting where tasks arriving dynamically need to be assigned to a set of servers where the probability of success of any job on any server is unknown. Using a stochastic multi-armed bandit setting, the authors showed that it is possible to achieve a proportionally fair allocation of jobs to servers in this setting. Multiple other works have studied the fairness problem in the stochastic multi-armed bandit setting [Patil et al., 2021, Joseph et al., 2016, Li et al., 2019, Chen et al., 2020]. There have also been some works recently that study algorithms with group fairness guarantees where we are concerned about being fair to a group of individuals (for example, a group might correspond to a race) instead of a single individual. [Freund et al., 2023] studies group fairness for algorithms used to assign refugees to different geographic locations. Baek and Farias [2021] studies the problem of sharing the cost of exploration in the stochastic multi-armed bandit model fairly among different groups.

Several prior works exist that design no-regret policies for specific non-additive functions. Blum and Burch [1997], Blum et al. [1999] used online learning techniques for solving the online paging and the Metrical Task System problem, which contains states. The notion of $\alpha$-fairness in the online setting has been considered in a couple of works including Si Salem et al. [2022] as explained previously in the introduction. Wang et al. [2022] considered an online resource allocation problem where the

objective is to guarantee a sublinear regret for the allocation efficiency and a sublinear minimum guarantee violation penalty. The authors proposed an online policy that achieves this goal by using a weighted $\alpha$-fair allocation on each round while sequentially tuning the weights and the exponents of the $\alpha$-fairness function. Although they use round-wise $\alpha$-fair allocation as a tool, their objective is not to optimize the $\alpha$-fairness of the cumulative allocation, which is the focus of our paper.

**Fairness in job scheduling:** Apart from the papers cited above, the fair resource allocation problem in the context of job scheduling has also been extensively investigated by the wireless communication community, predominantly under stochastic assumptions [Stolyar, 2005a, Kushner and Whiting, 2002]. Furthermore, contrary to the online learning setting, these works typically assume that the current reward vectors (*e.g.,* the channel states) are available to the scheduler before it makes scheduling decisions for each round. In this setting, the classic proportional fair scheduling algorithm emerges as a stochastic approximation-type gradient ascent algorithm for maximizing the logarithmic utility function [Stolyar, 2005a, Kelly et al., 1998]. Other notions of fair utility functions that have been studied in the context of wireless communication include max-min utility [Bertsekas and Gallager, 1991, Jaffe, 1981] and $\alpha$-fair utility [Mo and Walrand, 2000] (also called isoelastic utility function).

**Fairness in matching:** Fair matching algorithms have been studied in the context of resource allocation in online marketplaces in Bateni et al. [2021]. In this setting, a platform must dynamically allocate goods arriving at a platform in an online manner to customers who have their own utility and budget for each of these goods. The allocation of goods to customers must be fair to the customers while also maximizing the revenue for the platform. Different problems like online advertising where impressions must be served in an online manner to advertisers on a platform can be modeled under this setting. A proportionally fair stochastic approximation scheme was proposed in this paper. An algorithm with sublinear regret in a similar setting was proposed in Balseiro et al. [2021] where the fairness criteria were modeled as a regularizer for the objective function. Deng et al. [2023] studied how to incorporate machine-learned advice to improve fairness for bidders in the context of algorithmic bidding in online advertising. Competitive algorithms for optimizng both group and individual fairness in *online matching markets* which includes recommendation engines, crowdsourcing platforms etc. was proposed in Ma et al. [2023]. Fair matching algorithms for other domains like organ allocation Bertsimas et al. [2013] have also been proposed.

# B    Additional Examples

Beyond online fair caching, the `NOFRA` problem can capture a fair version of several other problems in the literature. In the following, we concretely present online fair job scheduling and online fair matching problems.

**Example B.1** (Online Job Scheduling [Even-Dar et al., 2009])**.** In this problem, there are $m$ machines that play the role of agents. A single job arrives at each round. The reward accrued by assigning the incoming job at round $t$ to the $i^{\text{th}}$ machine is given by $x_i(t)$ where $x_i(t) \in [\delta, 1], \forall i \in [m]$, where $\delta > 0$ is a small positive constant. Before the rewards for round $t$ are revealed, an online allocation policy selects a probability distribution $\boldsymbol{y}_t$ on $m$ machines such that

$$\sum_{i=1}^{m} y_i(t) = 1, \ y_i(t) \geq 0, \forall i \in [m].$$

The policy allocates a fraction $y_i(t)$ of the job to the $i^{\text{th}}$ machine $\forall i \in [m]$. As a result, the $i^{\text{th}}$ machine accrues a reward of $x_i(t)y_i(t)$ on round $t$. Hence, the cumulative reward accrued by the $i^{\text{th}}$ machine in a time-horizon of length $T$ is given by:

$$R_i(T) = 1 + \sum_{t=1}^{T} x_i(t)y_i(t), \ \forall i \in [m].$$

The objective of the online fair job scheduling problem is to design an online allocation policy that achieves a sublinear regret with respect to the $\alpha$-fairness of the cumulative rewards defined in (4). From the above description, it is immediately clear that this problem is a special case of the general `NOFRA` problem. The online job scheduling problem occurs in many practical settings, *e.g.,* in the targeted ad-campaigning problem discussed in the introduction, the ads can be modelled as jobs, and different groups of users can be modelled as machines. In OFDMA wireless systems, the

fair allocation of frequency resource blocks to different users where the wireless channels change dynamically can be straightforwardly modelled as an instance of the online job scheduling problem.

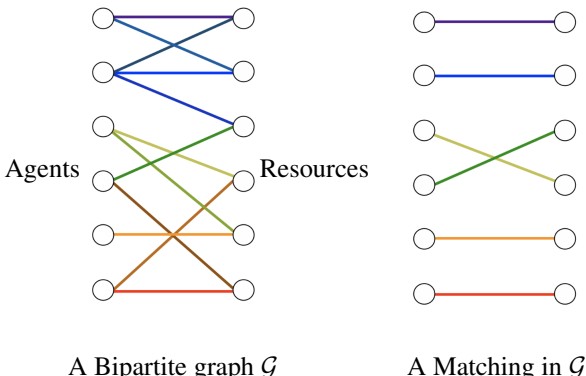

A Bipartite graph $\mathcal{G}$          A Matching in $\mathcal{G}$

Figure 4: Illustrating the Online Fair Matching (OFM) problem for zero-one demands. The existence of an edge denotes unit demand, and likewise, the absence of any edge implies zero demand.

**Example B.2** (Online Matching). Consider an $m \times m$ bipartite graph where the vertices on the left denote the agents and the vertices on the right denote the resources. On every round, each agent can be matched with one resource only, where we allow fractional matchings. The $m \times 1$ demand vector of the agent $i$ on round $t$ is denoted by $x_i(t)$. The $j^{\text{th}}$ component of the demand vector $x_i(t)$ denotes the potential reward accrued by the $i^{\text{th}}$ agent had it been completely matched with the $j^{\text{th}}$ resource on round $t$. Let the binary action variable $y_{ij}(t) \in [0, 1]$ denote the amount by which the agent $i$ is matched with resource $j$ on round $t$. Hence, the reward accrued by the agent $i$ on time $t$ is given by $\langle x_i(t), y_i(t) \rangle$. Let $\Delta$ denote the convex hull of all matchings. It is well known that $\Delta$ can be succinctly represented by the set of all $m \times m$ doubly stochastic matrices, *a.k.a.* the Birkhoff polytope [Ziegler, 2012]). In other words, in the OFM problem, the set of all feasible actions $\Delta$ consists of all $m \times m$ matrices $(y_{ij})_{i,j}$ satisfying the following constraints [8]:

$$\sum_{i=1}^{m} y_{ij} = 1, \ \forall j, \ \ \sum_{j=1}^{m} y_{ij} = 1, \ \forall i, \ \ 0 \le y_{ij} \le 1, \ \forall i, j. \tag{13}$$

Let the variable $R_i(t)$ denote the cumulative rewards accrued by the $i^{\text{th}}$ agent up to round $t$. Clearly,

$$R_i(t) = R_i(t-1) + \langle x_i(t), y_i(t) \rangle, \ R_i(0) = 1, \ \forall i \in [m], \tag{14}$$

It can be verified that Assumption 2 holds in this problem with $\mu = m^{-1}$ by noting that $m^{-1}\mathbf{1}_{m \times m} \in \Delta$. The objective of the Online Fair Matching (OFM) problem is to design an online matching policy that minimizes the $c$-Regret (4). From the above formulation, it can be immediately seen that the Online Matching problem is an instance of the NOFRA problem. Furthermore, it also generalizes the online scheduling problem described in Example B.1.

The online matching problem arises in numerous practical settings. For example, in the problem of online Ad Allocation, there are $m$ different advertisers whose ads need to be placed in $m$ different display slots on a webpage. Each slot can accommodate only one ad. On round $t$, a new user arrives and presents a reward vector $\boldsymbol{x}_i(t)$ for each advertiser. For example, the component $x_{ij}(t)$ could denote potential revenue accrued by the advertiser if the ad $i$ is placed on the $j^{\text{th}}$ slot on round $t$. The objective of the allocation policy is to match the ads to the slots on each round so that the total earned revenue is fairly distributed among the advertisers. Similar problems arise in designing recommendation systems for crowd-sourcing, online dating websites [Tu et al., 2014], and fair channel assignment in wireless networks [Altman et al., 2010].

---

[8]As before the fractional matching can be converted to a randomized integral matching via sampling. See Section C.6 for details.

## C  Proofs

### C.1  Upper bound on the diameter of the admissible sets

**Lemma 3.** *For the online shared caching problem the diameter of the admissible action set can be bounded as follows:*

$$Diam(\Delta_k^N) \leq \sqrt{2k}.$$

*Proof.* Let $x, y \in \Delta_k^N$. We have

$$\|x - y\|_2^2 = \sum_{i=1}^N (x_i - y_i)^2 = \sum_{i=1}^N |x_i - y_i||x_i - y_i| \overset{(a)}{\leq} \sum_{i=1}^N |x_i - y_i| \overset{(b)}{\leq} \sum_{i=1}^N |x_i| + \sum_{i=1}^N |y_i| \overset{(c)}{=} 2k,$$

where, in (a), we have used the fact that $0 \leq x_i, y_i \leq 1$, in (b), we have used triangle inequality, and in (c), we have used the fact that the sum of the components of each feasible vector is $k$.  □

**Lemma 4.** *For the online matching problem, the diameter of the admissible action set can be bounded as follows:*

$$Diam(\Delta) \leq \sqrt{2m}.$$

Let $x, y \in \Delta$, where $\Delta$ is the feasible set for the OFM problem. We have

$$\|x - y\|_2^2 = \sum_{i,j} (x_{ij} - y_{ij})^2 \leq \sum_{i,j} |x_{ij} - y_{ij}| \leq \sum_{ij} x_{ij} + \sum_{ij} y_{ij} \leq 2m,$$

where the inequalities follow from similar arguments as in the proof of the previous lemma.

### C.2  Proof of Lemma 1

*Proof.* The expression for $\text{Regret}_T(\beta^{1-\alpha})$ from Eqn. (9) can be split into the difference of two terms $A$ and $B$ as follows:

$$\text{Regret}_T(\beta^{1-\alpha}) = \beta^{-\alpha}\Big[ \underbrace{\sum_i \phi'(R_i(T)) \sum_{t=1}^T \langle x_i(t), y_i^* \rangle}_{(A)} - \beta \underbrace{\sum_i \phi'(R_i(T)) \sum_{t=1}^T \langle x_i(t), y_i(t) \rangle}_{(B)} \Big].$$

Also, denote the corresponding terms in the regret expression (10) for the surrogate learning problem by $A'$ and $B'$. We will now separately bound each of the above two terms in terms of the corresponding terms in the regret expression (10) for the surrogate learning problem.

**Proving $A \leq A'$:**  Recall that the utility function $\phi(\cdot)$ is concave. Hence, its derivative is non-increasing. Furthermore, under the action of any policy, the cumulative reward $R_i(\cdot)$ is non-decreasing for each user $i \in [m]$. Thus, we have $\phi'(R_i(t-1)) \geq \phi'(R_i(T))$ for all $t \in [T], i \in [m]$. Hence,

$$A \leq \sum_{t=1}^T \langle \sum_i \phi'(R_i(t-1))x_i(t), y_i^* \rangle \leq A'. \tag{15}$$

**Proving** $B' \le (1-\alpha)^{-1}(B+m)$**:** We have

$$
\begin{aligned}
B' &= \sum_i \sum_{t=1}^{T} \phi'(R_i(t-1))\langle x_i(t), y_i(t)\rangle \\
&= \sum_i \sum_{t=1}^{T} \phi'(R_i(t-1))\big(R_i(t) - R_i(t-1)\big) \\
&\stackrel{(a)}{\le} \sum_i \int_0^{R_i(T)} \phi'(R)dR \\
&= \sum_i \phi(R_i(T)) \\
&\stackrel{(b)}{=} (1-\alpha)^{-1} \sum_i \phi'(R_i(T))R_i(T) \\
&\stackrel{(c)}{=} (1-\alpha)^{-1} \sum_i \phi'(R_i(T))(\sum_{t=1}^{T}\langle x_i(t), y_i(t)\rangle + 1) \\
&\le (1-\alpha)^{-1}(B+m), \quad\quad\quad\quad\quad\quad\quad\quad\quad\quad (16)
\end{aligned}
$$

where, in (a), we have used the fact that the derivative $\phi'(\cdot)$ is non-increasing and $R_i(t) - R_i(t-1) = \langle x_i(t), y_i(t)\rangle \le 1$ (see Appendix C.3 below for a proof by picture), in (b), we have used the explicit form of the $\alpha$-fair utility function to substitute $x\phi'(x) = (1-\alpha)\phi(x)$, and in (c), we have used (2). Combining (15) and (16) and choosing $\beta = (1-\alpha)^{-1}$, we conclude that, letting L-Regret$_T$ be the surrogate regret defined as in (10),

$$
\text{Regret}_T((1-\alpha)^{-(1-\alpha)}) \le (1-\alpha)^{\alpha} \cdot \text{L-Regret}_T + (1-\alpha)^{-(1-\alpha)}m.
$$

$\square$

### C.3 Graphical proof of the inequality:

$$
\phi'(R_i(t-1))\big(R_i(t) - R_i(t-1)\big) \le \int_{R_i(t-1)-1}^{R_i(t)-1} \phi'(R)dR. \quad\quad\quad (17)
$$

*Proof.* We use the fact that $\phi'(\cdot)$ is a non-increasing function and $0 \le R_i(t) - R_i(t-1) \le 1$. Then the inequality (17) follows straightforwardly from the schematic below.

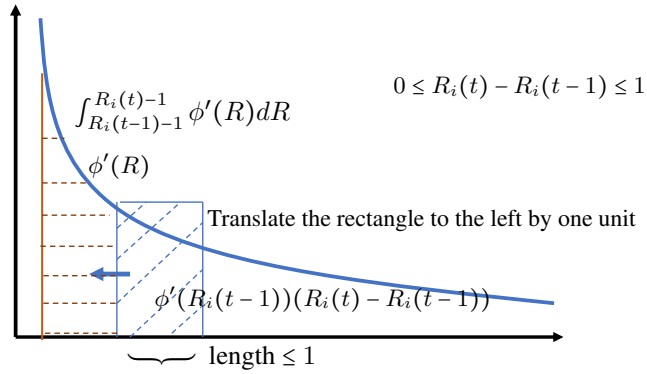

Figure 5: Graphical proof of (17)

The final inequality (a) then follows by summing up the above inequalities. $\square$

### C.4 Proof of Lemma 2

*Proof.* We will be using the following adaptive regret bound for the online gradient descent policy with an appropriate adaptive step sizes sequence. We will see that the norm of the gradients diminishes

at a steady rate under the action of the `OFA` policy. Hence, the data-dependent regret bound plays a central role in a tight regret analysis of the `OFA` policy.

**Theorem 3** (Theorem 4.14 of Orabona [2020]). *Let $\Delta \subset \mathbb{R}^d$ be a convex set with a diameter $D$. Let us consider a sequence of linear reward functions with gradients $\{g_t\}_{t \geq 1}$. Run the Online Gradient Ascent policy with step sizes $\eta_t = \frac{D}{\sqrt{2 \sum_{\tau=1}^t \|g_\tau\|_2^2}}, 1 \leq t \leq T$. Then the standard regret under the OGA policy can be upper-bounded as follows:*

$$Regret_T \leq D\sqrt{2\sum_{t=1}^T \|g_t\|_2^2}. \tag{18}$$

Note that the gradient component $g_i$ corresponding to the $i^{\text{th}}$ user for the surrogate problem (10) on round $t$ is given by the vector $g_i(t) = \phi'(R_i(t))x_i(t)$. Using the above data-dependent static regret bound (18), the regret achieved by the OGA policy for the surrogate problem for any round $T \geq 1$ can be upper-bounded as follows:

$$\text{L-Regret}_T = O\left(\sqrt{\sum_{t=1}^T \sum_i \phi'(R_i(t))^2}\right) = O\left(\sqrt{\sum_{t=1}^T \sum_i \frac{1}{R_i(t)^{2\alpha}}}\right), \tag{19}$$

where we have used the fact that the demand vectors at each round are bounded.

Clearly, the regret bound (19) depends on the sequence of the cumulative rewards $\{\boldsymbol{R}(t)\}_{t \geq 1}$, which is implicitly controlled by the past actions of the online policy itself. By the definition of regret, for any fixed allocation $y^* \in \Delta_k^N$, we have for any time step $T$:

$$\sum_{t=1}^T \sum_i \langle \phi'(R_i(t-1))x_i(t), y_i(t) \rangle \geq \sum_{t=1}^T \sum_i \langle \phi'(R_i(t-1))x_i(t), y_i^* \rangle - \text{L-Regret}_T, \tag{20}$$

where $\text{L-Regret}_T$ denotes the worst-case cumulative regret of the adaptive OGD policy up to time $T$. Since the cumulative reward of each user is monotonically non-decreasing, we have:

$$R_i(T) \geq 1, \forall i, \forall T. \tag{21}$$

Substituting this bound to the regret bound in (19), we obtain our first (loose) upper-bound for the regret of the fair allocation problem (10):

$$\text{L-Regret}_T = O(\sqrt{T}). \tag{22}$$

Note that this bound might be too loose as the offline benchmark itself could be smaller in magnitude than this regret bound. A closer inspection reveals the root cause for this looseness - the cumulative reward lower bound (21) is too loose for our purpose, as cumulative rewards grow steadily with time, depending on the online policy. We now strengthen the above upper-bound using a novel *bootstrapping* method, that *simultaneously* tightens the lower bound for the cumulative rewards (21) and, consequently, improves the regret upper bound (19).

Using the fact that $\langle x_i(t), y_i(t) \rangle = R_i(t) - R_i(t-1)$, and following the same calculations up to the fourth step of (16), we have

$$\sum_i \sum_{t=1}^T \phi'(R_i(t-1))\langle x_i(t), y_i(t) \rangle \leq \sum_i \phi(R_i(T)). \tag{23}$$

Furthermore, lower bounding $\phi'(R_i(t-1))$ by $\phi'(R_i(T))$, we have

$$\sum_{t=1}^T \sum_i \langle \phi'(R_i(t-1))x_i(t), y_i^* \rangle \geq \sum_i \phi'(R_i(T))R_i^*(T), \tag{24}$$

where $R_i^*(T) \equiv \sum_{t=1}^T \langle x_i(t), y_i^* \rangle$ is the cumulative reward accrued by a fixed allocation $y^* \in \Delta$ up to time $T$. Using Assumptions 1 and 2 and choosing $y_i^* = \mu \mathbb{1}_N$, we have $R_i^*(T) \geq \mu\delta T, \forall i$. Hence, combining (23), (24), and (20), we have

$$\sum_i \phi(R_i(T)) \geq \mu\delta T \sum_i \phi'(R_i(T)) - \text{L-Regret}_T. \tag{25}$$

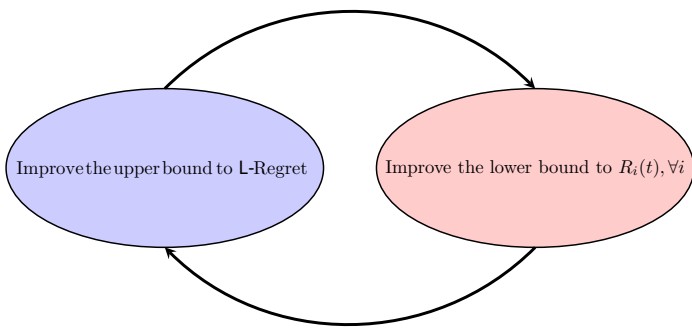

Figure 6: Illustrating the new *Bootstrapping* technique used in the proof of Lemma 2

Since $0 < R_i(T) \le T, \forall i$, and $\phi(\cdot)$ is monotone non-decreasing, for any user $i \in [m]$, the above inequality yields:

$$\frac{m}{(1-\alpha)T^\alpha} \ge \frac{\mu\delta}{R_i^\alpha(T)} - \frac{\text{L-Regret}_T}{T}.$$

*i.e.,*

$$R_i^\alpha(T) \ge \mu\delta\left(\frac{\text{L-Regret}_T}{T} + \frac{m}{1-\alpha}T^{-\alpha}\right)^{-1} = \Omega(T^{\min(1/2,\alpha)}), \forall i \in [m], \tag{26}$$

where we have used our previous upper bound (22) on the regret. Eqn. (26) is our key equation for carrying out the bootstrapping process as it lower bounds the minimum cumulative reward of users in terms of the worst-case regret. Now we consider two cases:

**Case-I:** $0 \le \alpha \le 1/2$

In this case, we immediately have $R_i(T) = \Omega(T), \forall i, T$. Substituting this bound on (19), we have

$$\text{L-Regret}_T = \begin{cases} O(\sqrt{\log T}) & \text{if } \alpha = 1/2 \\ O(T^{1/2-\alpha}) & \text{if } 0 < \alpha < 1/2. \end{cases}$$

**Case-II:** $1/2 < \alpha < 1$

In this case, using the bound (26), we have $R_i(T) = \Omega(T^{1/2\alpha}), \forall i, T$. Using the regret bound (19), the regret can be bounded as

$$\text{L-Regret}_T = O\left(\sqrt{\sum_{t=1}^{T}\frac{1}{t}}\right) = O(\sqrt{\log T}).$$

Substituting this bound again in (26), we have $R_i(T) \ge \Omega(T)$. This, in turn, yields the following regret bound:

$$\text{L-Regret}_T = O\left(\sqrt{\sum_{t=1}^{T}\frac{1}{t^{2\alpha}}}\right) = O(1).$$

$\square$

## C.5 Simplified Pseudocode for the Online Fair Allocation Policy for the Caching Problem

---

**Algorithm 2** Online Fair Allocation Caching (`OPFC`)

---

1: **Input:** Fairness parameter $0 \le \alpha < 1$. Online file request sequence $\{x_i(t)\}_{t=1}^T$ from user $i, i \in [m]$, Euclidean projection oracle $\Pi_{\Delta_k^N}(\cdot)$ onto the feasible convex set $\Delta_k^N$.
2: **Output:** Sequence of file inclusion probabilities $\{y_t\}_{t=1}^T$ and samples from this distribution
3: $R_i \leftarrow 1, \forall i \in [m], S \leftarrow 0.$          ▷ *Initialization*
4: **for each** round $t = 2 : T :$ **do**
5:      $g \leftarrow \sum_{i=1}^m \frac{x_i(t-1)}{R_i^\alpha}$          ▷ *Computing the gradient*
6:      $S \leftarrow S + \|g\|_2^2,$          ▷ *Accumulating the norm of the gradients*
7:      $y \leftarrow \Pi_{\Delta_k^N}\left(y + \frac{k}{\sqrt{S}}g\right)$          ▷ *Updating the inclusion probabilities using OGA*
8:      The users reveal their file requests for round $t$ $\{x_i(t), i \in [m]\}$.
9:      $R_i \leftarrow R_i + \langle x_i(t), y \rangle, \ \forall i \in [m].$          ▷ *Updating cumulative hits*
10:      Sample a subset of $k$ files by invoking Algorithm 3
11: **end for each**

---

## C.6 High-probability regret bound for randomized integral allocations

Theorem 2 establishes $c_\alpha \equiv (1-\alpha)^{-(1-\alpha)}$-regret guarantee for the Online Fair Allocation (`OFA`) policy for fractional allocations where the set of admissible actions $\Delta$ is the convex hull of integral allocations. In the following, we show that a sub-linear $c_\alpha$-regret guarantee holds with high probability when randomized integral actions are chosen according to the optional step (9) in the `OFA` policy. Recall that for integral allocation, on each round $t$ we independently sample an integral allocation $Y_t \in \{0,1\}^{mN}$, that matches with the fractional allocation $y_t$ in expectation, *i.e.,* $\mathbb{E}[Y_t] = y_t$ where $y_t$ is the fractional allocation vector recommended by the `OFA` policy. A fractional allocation vector $y_t$ in the convex hull of integral allocations can be turned into a randomized integral allocation by expressing $y_t$ as a convex combination of integral allocations and randomly sampling one of the integral allocations with appropriate probabilities. As Appendix C.7 shows, for many problems with structure, such a decomposition can be efficiently computed.

**Regret Analysis:** By appealing to [Cesa-Bianchi and Lugosi, 2006, Lemma 4.1], it is enough to consider an oblivious adversary that fixes the sequence of demand vectors $\{x(t)\}_{t \ge 1}$ before the game commences. Let the random variable $\mathsf{R}_i(T)$ denote the random cumulative reward obtained by the $i^{\text{th}}$ user under the randomized integral allocation policy and the deterministic variable $R_i(T)$ is defined as in Eqn. (2). By construction, we have $\mathbb{E}[\mathsf{R}_i(T)] = R_i(T)$. Furthermore, since $\mathsf{R}_i(T)$ is the sum of $T$ independent random variables, each of magnitude at most one, from the standard Hoeffding's inequality, we have

$$\mathbb{P}(|\mathsf{R}_i(T) - R_i(T)| \ge \lambda) \le \exp(-2\lambda^2/T), \forall i \in [m].$$

Hence, with probability at least $1 - \texttt{poly}(1/T)$, the aggregate $\alpha$-fair utility (3) accrued by the randomized policy can be lower bounded as:

$$
\begin{aligned}
(1-\alpha)^{-1}\sum_i \mathsf{R}_i^{1-\alpha}(T) &\ge (1-\alpha)^{-1}\sum_i \left(R_i(T) - O(\sqrt{T\log T})\right)^{1-\alpha} \\
&\overset{(a)}{\ge} (1-\alpha)^{-1}\sum_i R_i^{1-\alpha}(T) - O\left((T\log T)^{\frac{1-\alpha}{2}}\right),
\end{aligned}
\tag{27}
$$

where in inequality (a), we have used the fact that $(x+y)^{1-\alpha} \le x^{1-\alpha} + y^{1-\alpha}, \forall 0 \le \alpha \le 1, x, y \ge 0$. Combining the above with the approximate regret bound in Theorem 1, and taking the dominant term, we conclude that the $c_\alpha$-regret for the randomized integral allocation policy is upper-bounded by $O\left((T\log T)^{\frac{1-\alpha}{2}}\right)$ w.h.p. for all $0 \le \alpha \le 1$. Unfortunately, unlike Theorem 1, the integral allocation policy incurs a non-trivial (but sublinear) $c_\alpha$-regret for the entire range of the fairness parameter $0 \le \alpha < 1$.

## C.7 Efficiently sampling an integral allocation from a mixed allocation in the convex hull

**1. Online shared caching:** For the online shared caching problem, the admissible set $\Delta$ is given by Eq. (5). Clearly, incidence vectors of all $k$-sets, containing $k$ 1's and $N - k$ zeros, belong to $\Delta$. Furthermore, given any vector in $\boldsymbol{y} \in \Delta$, a randomized integral allocation can be sampled in linear time using Madow's sampling scheme, described in Algorithm 3, which yields the vector $\boldsymbol{y}$ in expectation.

---

**Algorithm 3** MADOW-SAMPLE $(p)$

---

1: **Input:** A universe $[N]$ of size $N$, the cardinality of the sampled set $k$, and a marginal inclusion probability vector $p \in \Delta_k^N$.
2: **Output:** A random $k$-set $S$ with $|S| = k$ such that, $\mathbb{P}(i \in S) = p_i, \forall i \in [N]$
3: Define $\Pi_0 = 0$, and $\Pi_i = \Pi_{i-1} + p_i, \forall 1 \leq i \leq N$.
4: Sample a uniformly distributed random variable $U$ from the interval $[0, 1]$.
5: $S \leftarrow \varnothing$
6: **for each** $i \leftarrow 0$ to $k - 1$ **do**
7:     Select the element $j$ if $\Pi_{j-1} \leq U + i < \Pi_j$.
8: **end for each**
9: **return** $S$

---

**2. Online job scheduling:** In the online job scheduling problem, the admissible action set $\Delta$ is given by the $N$-simplex. Given any point on the $N$-simplex, it is trivial to randomly sample a coordinate using a single uniform random variable.

**3. Online matching:** In the online matching problem, the admissible action set $\Delta$ is given by the Birkhoff polytope (13). Given any point $\boldsymbol{y}$ in the Birkhoff polytope, it can be efficiently decomposed as a convex combination of a small number of matchings using the Birkhoff-von-Neumann (BvN) decomposition algorithm [Valls et al., 2021]. Using the decomposition coefficients, a matching can be randomly sampled that exactly matches the point $\boldsymbol{y}$ in expectation.

## C.8 Non-Convexity of the optimal offline benchmark for the online job scheduling problem

Consider the job scheduling problem in the reward maximization setting as described in Example B.1. Clearly, the $\alpha$-fairness metric accumulated by the online policy is concave in the regime $0 \leq \alpha \leq 1$. However, the following proposition shows that the offline static benchmark for this problem is *non-convex* for the $\alpha$-fair utility function in the regime $0 < \alpha < 1$.

> **Proposition 1.** *The offline optimal $\alpha$-fair reward function for the job scheduling problem is non-convex for $0 < \alpha < 1$.*

*Proof.* Let the probability distribution $\boldsymbol{y}^*$ be an optimal static offline allocation vector for the reward sequence $\{\boldsymbol{x}_t\}_{t \geq 1}$. Also, let $R_T(i) \equiv \sum_t x_i(t)$ be the cumulative reward observed by the $i^{\text{th}}$ machine, $1 \leq i \leq m$. Then the optimal allocation $\boldsymbol{y}^*$ maximizes the following objective:

$$(1 - \alpha)^{-1} \sum_i (y_i R_T(i))^{1-\alpha}. \tag{28}$$

s.t. the constraints $\sum_i y_i = 1, \boldsymbol{y} \geq \boldsymbol{0}$. Since $0 < \alpha < 1$, the objective function is strongly concave and the optimal solution $\boldsymbol{y}^*$ is unique. Using the standard Hölder's inequality with the conjugate norms $p = 1/1-\alpha, q = 1/\alpha$, we can upper bound the objective (28) as

$$
\begin{aligned}
(1 - \alpha)^{-1} \sum_{i=1}^m (y_i R_T(i))^{1-\alpha} &\leq (1 - \alpha)^{-1} \left(\sum_i y_i\right)^{1-\alpha} \left(\sum_i R_T(i)^{\frac{1-\alpha}{\alpha}}\right)^\alpha \\
&= (1 - \alpha)^{-1} \left(\sum_i R_T(i)^{\frac{1-\alpha}{\alpha}}\right)^\alpha, \tag{29}
\end{aligned}
$$

where we have used the constraint $\sum_i y_i = 1$ in the last equality. The upper-bound is achieved by the following distribution

$$y_i^* = \frac{R_i^{\frac{1-\alpha}{\alpha}}}{\sum_i R_i^{\frac{1-\alpha}{\alpha}}}, \quad \forall i \in [m]. \tag{30}$$

Hence, the optimal offline reward (28) is given by

$$\mathcal{R}^* = (1 - \alpha)^{-1} \left( \sum_i R_T(i)^{\frac{1-\alpha}{\alpha}} \right)^\alpha. \tag{31}$$

To show that $\mathcal{R}^*$ is non-convex in the cumulative reward vector $\boldsymbol{R}_T$ in the regime $0 < \alpha < 1$, consider the case of two users. Letting $x = R_T(1), y = R_T(2)$ and ignoring the positive pre-factor, the function under consideration is given as follows:

$$f(x,y) = \left( x^{\frac{1-\alpha}{\alpha}} + y^{\frac{1-\alpha}{\alpha}} \right)^\alpha. \tag{32}$$

Recall that a function is convex if and only if the determinants of all leading principal minors of its Hessian matrix are non-negative. A straightforward computation yields the following expression for the determinant of the Hessian of the function $f(x,y)$:

$$\det(\nabla^2 f(x,y)) = (2\alpha - 1) \frac{(\alpha - 1)^2 x^{1/\alpha - 1} y^{1/\alpha - 1} (x^{1/\alpha - 1} + y^{1/\alpha - 1})^{2\alpha}}{(yx^{1/\alpha} + xy^{1/\alpha})^2}.$$

The determinant becomes strictly negative in the regime $0 < \alpha < \frac{1}{2}$. This shows that the function (32) is non-convex for $0 < \alpha < \frac{1}{2}$. On the other hand, we have

$$\frac{\partial^2 f(x,y)}{\partial x^2} = (\alpha - 1) \frac{yx^{1/\alpha - 2}(x^{1/\alpha - 1} + y^{1/\alpha - 1})^\alpha (\alpha^2 yx^{1/\alpha} + 2\alpha xy^{1/\alpha} - xy^{1/\alpha})}{\alpha(yx^{1/\alpha} + xy^{1/\alpha})^2}.$$

In particular, at the point $(1,1)$ the above second partial derivative evaluates to be

$$\frac{\partial^2 f(x,y)}{\partial x^2}\Big|_{(x,y)=(1,1)} = -\frac{2^{\alpha-2}(1-\alpha)}{\alpha}(\alpha^2 + 2\alpha - 1),$$

which is strictly negative for $\sqrt{2} - 1 < \alpha < 1$. Taking the above two results together, we conclude that the offline optimal reward function (32) is non-convex for $0 < \alpha < 1$. $\qquad\square$

## C.9 Proof of Theorem 2 (lower bounding the approximation factor)

Consider the following instance of the online shared caching problem with $m = 2$ users and a cache of unit capacity. Assume that the library's size $(N)$ is sufficiently large. Let the total length of the request sequence be $T$ rounds and let $\eta \in [0,1]$ be some constant fraction to be fixed later. Consider two different file request sequences:

- **Instance 1:** For the first $\eta T$ rounds, user 1 always requests file 1 and user 2 always requests file 2. For the next $(1 - \eta)T$ rounds, user 1 always requests file 2 and user 2 requests a file chosen uniformly at random from the library.

- **Instance 2:** For the first $\eta T$ rounds, as in Instance 1, user 1 always requests file 1 and user 2 always requests file 2. However, for the next $(1 - \eta)T$ rounds, user 1 requests a random file chosen uniformly at random from the library and user 2 always requests file 1.

We now lower bound the approximation factor achievable by any online policy for the above two request sequences.

**Optimal static offline policy:** It is easy to see that the optimal offline strategy is to cache file 2 for all $T$ rounds. Hence, the total $\alpha$-fairness objective accrued by the static offline policy is

$$\text{Offline } \alpha\text{-fairness} = (1 - \alpha)^{-1} T^{1-\alpha} \left[ \eta^{1-\alpha} + (1 - \eta)^{1-\alpha} \right].$$

Clearly, the same fairness objective is achieved for the second instance by a static policy that always caches file 1.

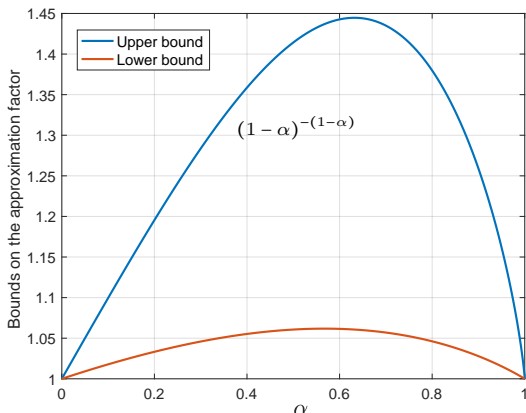

$(1-\alpha)^{-(1-\alpha)}$

Figure 7: Comparison between the upper and lower bounds of the approximation ratio

**Online Policy:** Suppose that during the first $\eta T$ rounds, the online policy caches file 1 for $\gamma$ fraction of the time and file 2 for $1 - \gamma$ fraction of the time. Since the policy is online, the fraction $\gamma$ remains the same for both instances. Clearly, for the next $(1 - \eta)T$ time slots, an optimal online policy caches file 2 for instance 1 and file 1 for instance 2. Taking the worst of the two instances, any online policy achieves the following fairness objective:

$$\text{Online } \alpha\text{-fairness} = (1 - \alpha)^{-1} T^{1-\alpha} \min\big(g(\gamma), g(1 - \gamma)\big),$$

where $g(\gamma) \equiv \big(\gamma\eta + (1 - \eta)\big)^{1-\alpha} + \big((1 - \gamma)\eta\big)^{1-\alpha}$. From simple calculus, it follows that the function $g(\gamma)$ is non-increasing for $0 \leq \eta \leq 1/2, 0 \leq \alpha < 1$. This implies that the online $\alpha$-fairness is maximized when $\gamma^* = \frac{1}{2}$. Hence, the $\alpha$-fairness metric achieved by any online policy for the worst of the above two instances is upper bounded by:

$$\text{Online } \alpha\text{-fairness} \leq \big(1 - \eta/2\big)^{1-\alpha} + \big(\eta/2\big)^{1-\alpha}.$$

Hence, the achievable approximation ratio $c_\alpha$ for any online policy with a sublinear $c_\alpha$-regret is lower bounded as follows:

$$\text{Approx. ratio} \geq \frac{\text{Offline } \alpha\text{-fairness}}{\text{Online } \alpha\text{-fairness}} = \frac{\eta^{1-\alpha} + (1 - \eta)^{1-\alpha}}{(1 - \eta/2)^{1-\alpha} + (\eta/2)^{1-\alpha}}.$$

Taking the maximum of the RHS with respect to the parameter $\eta$ yields the desired result.

## D  Additional Numerical Experiments

In this section, we provide a more detailed description of how the CDN dataset was preprocessed and additional numerical experiments on fair scheduling.

### D.1  Preprocessing the CDN dataset

We give additional details regarding preprocessing the CDN dataset for the experiments here. As explained in section 4 we set $T = 400$ rounds, number of users $m = 4$, cache size $C = 10$ and, library size $N = 50$. For preprocessing, first, we sort the data according to the timestamp and discard all the files with an ID greater than $N$, and then only consider the first $m \times T = 1600$ requests among the remaining requests. We then relabel the files from 1 to $N$ in descending order of the number of requests such that among these 1600 requests, file 1 is the most requested file, followed by the second file, and so on. We then assign the requests sequentially to users in decreasing order of priority from users 1-4, i.e., we first try to assign a request to user 1, failing which we try to assign it to user 2, and then to users 3 and 4. We can only assign a request to a user if the total number of requests assigned to that user is less than or equal to $T = 400$ and the following conditions are met for users 1-3: (1)

user 1: file ID between 1-6; (2) user 2: file ID between 1-9; (3) user 3: file ID between 1-12. User 4 can be assigned any file. Thus, we allocate files to users in decreasing order of their popularity, such that user 1 tends to request common files, while user 4 tends to request less common files. This skews the file request distribution, making a fair cache allocation for all users difficult.

## D.2 Fair Scheduling

**Dataset:** The datasets are generated by simulating a wireless channel. For the first dataset (*WLAN*), the simulator generates channel gains and shadowing values for each user, computes the received power and noise for each user, adds Rayleigh fading to the signal-to-noise ratios (SNRs), and then normalizes the SNRs by the maximum value to generate rewards. We take the horizon $T = 5000$ and number of users $m = 5$. The second dataset (*MCS*) is generated using Modulation and Coding Scheme table which defines the data rates for each user depending on the radio link quality. Here, the number of users is $m = 2$ and the horizon is at $T = 2000$. Although regret guarantees for $\alpha > 1$ become vacuous, to clearly demonstrate the fairness characteristics of algorithms, we run simulations for $\alpha \in [0, 2]$.

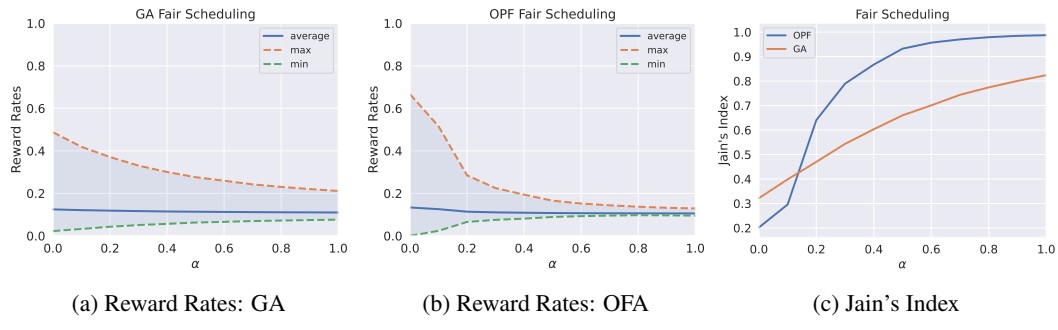

(a) Reward Rates: GA          (b) Reward Rates: OFA          (c) Jain's Index

Figure 8: Scheduling (WLAN)

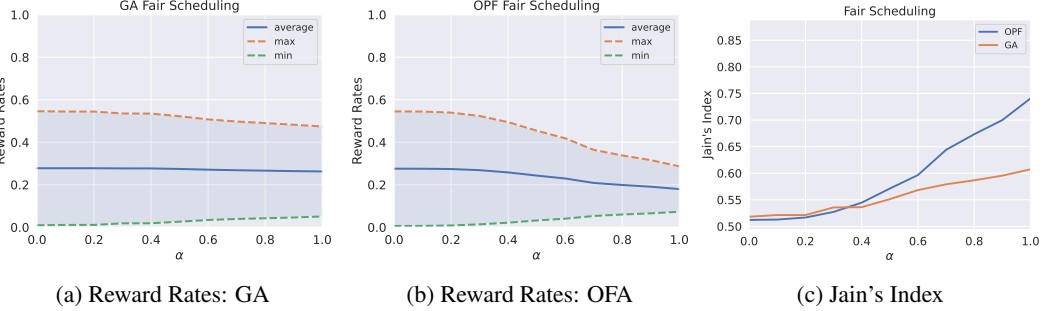

(a) Reward Rates: GA          (b) Reward Rates: OFA          (c) Jain's Index

Figure 9: Scheduling (WLAN)

**Comparison:** We compare our policy with the Gradient Algorithm (GA) described by Stolyar [2005b]. The Gradient Algorithm is a widely used in fairly scheduling transmission of multiple users via a time-varying wireless channel. Stolyar considers a model where a finite set of queues are served by a switch. The switch has a finite number of switch states $M$. The sequence of states at each time steps form an irreducible Markov chain. At each state $m$, a scheduling decision $k \in K(m)$ is made, where $K(m)$ is a finite set of decisions. Each scheduling decision has a corresponding vector of service rates given by $\mu^m(k)$. Let $V$ be the set of all long-term average service rate vectors. The goal is to have a scheduling policy that gives an average service rate solving the maximization problem

$$\max_{u \in V} H(u), \tag{33}$$

where $H(\cdot)$ is a strictly concave smooth utility function. The Gradient Algorithm makes a scheduling decision

$$k(t) \in \operatorname*{argmax}_{k} \langle \nabla H(X(t)), \mu^m(k) \rangle,$$

where $X(t)$ is updated as

$$X(t+1) = (1-\beta)X(t) + \beta\mu^m(k),$$

for some initial value of $X(0)$ and a fixed parameter $\beta > 0$. The Gradient Algorithm converges to an optimal $u^*$ that maximizes (33) as $\beta \to 0$.

Figures 8a, 8b, 9a and 9b compares the average, minimum and maximum reward rates of both the policies as $\alpha$ increases. The Jain's Index of both the policies is shown in Figures 8c and 9c. It can be clearly observed that for $\alpha \in [0, 1)$, the OFA policy performs better in terms of fairness, but for larger $\alpha$'s, the GA outperforms the OFA policy by a small margin. Recall that Stolyar only provides theoretical guarantees for a stochastic setting whereas we give a stronger regret bound in an adversarial setting at the expense of slight performance degradation in practice.

