# OpenReview forum: "No-regret Algorithms for Fair Resource Allocation"
_NeurIPS.cc/2023/Conference — NeurIPS 2023 poster_

### Official Review · Reviewer_wJNK · 2023-07-03

**Soundness:** 4 excellent
**Presentation:** 4 excellent
**Contribution:** 3 good
**Rating:** 7
**Confidence:** 3

**Summary:**

The paper studies an online allocation problem where the goal is to find \alpha-fair solutions. The authors provide an algorithm that provides a constant approximate regret to this problem, along with a non-tight lower bound that improves a previous known result. The suggested algorithm is compared with other approaches over synthetic and real data on the fair caching and  fair scheduling problems.

**Strengths:**

- Well-written paper

-Important problem with real life applications

-Technically sound and strong submission. Nice techniques are used.

**Weaknesses:**

-The upper and lower bounds are not tight.

-The lower bound that is provided is a slight improvement from a previous known lower bound which indicates that c_{\alpha} >1, as shown in Figure 7 in the appendix.

-The experimental results are not very enlightening. It would be nice if the authors highlight what we really learn from them, provide an intuitive explanation why for smaller values of \alpha their algorithm is outperformed from other algorithms with respect to the average case etc.

**Questions:**

Something is inconsistent with respect to the line 5 of Algorithm 1 and the description in lines 230-23. In my understanding, in round t, the agent divides x_i(t-1) with R_i(t-1). In the algorithm it seems that indeed you do that, but in the description, you are saying that essentially x_i(t-1) is divided by R_i(t-2). Which one is the correct?

Codes of the experiments are not included in the supplementary material!

**Limitations:**

Yes, the authors have adequately addressed the limitations of their work.

---

> ### Author Rebuttal · Authors · 2023-08-06
>
> We thank the reviewer very much for their feedback, which we address below.
>
> [$\textbf{On the experimental results}$]: The goal of our algorithm is to provide fairness (i.e., to roughly ensure that hit rates of all users are as close to each other as possible) without significantly sacrificing the average hit rate. As $\alpha$ is increased, the algorithm should tend towards prioritizing fairness over average hit rate. Our experiments demonstrate that the algorithm indeed succeeds in these goals. For both the synthetic dataset and CDN dataset, we see that our algorithm achieves comparable average hit rate to the algorithm of Si Salem et al. at small $\alpha$, while somewhat significantly outperforming on average hit rate for larger $\alpha$. At the same time, the algorithm outperforms on the fairness metrics for the full range of $\alpha$ on the CDN dataset. On the synthetic dataset, the two algorithms are comparable in terms of fairness — neither one consistently outperforms the other.
>
> Our algorithm consistently performs within the range of the offline optimal allocation — which is a theoretical baseline that cannot be run in practice (as it would require knowing the full sequence of requests ahead of time). This offline optimal solution gives an upper bound on the performance of any online algorithm. Finally, our algorithm is comparable or better in terms of average hit rate, but, as expected, stronger in terms of fairness metrics, as compared to algorithms like LRU and LFU which do not try to ensure fairness.
>
> Regarding performance for small $\alpha$: we disagree that our algorithm is consistently outperformed in this regime. It performs very similarly to the Si Salem et al. algorithm, and in fact outperforms the LRU and LFU baselines. It is only outperformed by the offline optimal solution, which as discussed is meant to give a theoretical upper bound on performance and cannot be implemented in the online setting.
>
>
> [$\textbf{Question on Algorithm 1}$]: In line 5, we indeed divide $x_i(t-1)$ by $R_i(t-1)^\alpha$. Thanks for pointing out this typo — we will correct the description in the revised version.
>
> [$\textbf{Releasing Codes}$] We will release the code upon acceptance of the paper.

---

> > ### Comment · Reviewer_wJNK · 2023-08-18
> >
> > Thank you for your response. I am convinced about the merit of the experiments.
> >
> > After reading the other reviews and the author response, I will revise my score from a 6 to a 7.

---

### Official Review · Reviewer_q9U9 · 2023-07-05

**Soundness:** 3 good
**Presentation:** 4 excellent
**Contribution:** 3 good
**Rating:** 7
**Confidence:** 4

**Summary:**

The paper studies a general online resource allocation in which there are $m$ agents and a limited resource to be allocated over T rounds.  The goal is to achieve sublinear regret for the aggregate utilities of the agents when compared to the optimal fixed offline allocation policy. In the online problem, at each round an agent has a demand vector; first, the decision-maker decides on an allocation according to some policy, and then the aggregate demand vector is revealed. An unrestricted adversary fixes the demand vector (this assumption departs from previous works). The utility for any agent is given by a concave utility function controlled by a parameter $\alpha$, which varies between 0 and 1 and expresses the trade-off between fairness and efficiency, going from a very equitable allocation to the classic problem of maximizing social welfare without looking at fairness.

The NOFRA problem is defined, along with mild assumptions on the demand and allocation vectors. A main challenge comes from the fact that the objective function, which is the sum of the agents’ utilities, is not additive and, thus, not separable across time. Instead, we have to optimize in a “global” way. The objective, as mentioned before, is to design an online resource allocation policy that minimizes regret.

The contributions of the paper include (1) an efficient online resource allocation policy called OFA (Online Fair Allocation) that achieves approximate sublinear regret for any given $\alpha$ in the utility functions, (2) a (non-tight) lower bound (that also improves over previous work) on the approximate factor needed to attain sublinear regret for any online learning policy, and, (3) numerical simulation results with several baselines for the fair caching problem.

**Strengths:**

- The treatment of the problem is fairly complete, and, despite not obtaining a tight bound, it is important that the authors establish that indeed some notion of approximate regret minimization is needed to achieve any sublinear bound over time.
- Technical side: A key step is lemma 1, which introduces a surrogate problem, whose regret upper bounds the regret of the original one, and it can be solved as an online linear optimization problem. This is an elegant way to overcome the fact that the objective is not additive over rounds. The introduction of the technique to control the norm of gradients is also nice.
- The comparison with previous work is clear, the improvements and technical innovations are also clear, so overall the authors do a good job in convincingly placing their work in the literature. I also find quite interesting the transitions in the regret bound for different values of $\alpha$. It is also positive that it recovers the known regret bound when $\alpha=0$.


**Weaknesses:**

- Some of the experiments are not very convincing to make the theory stand out more. For example, assuming that the average hitrate is what we would consider more important, there’s not any improvement (in the real-world dataset it’s actually performing worse) compared to the Si Salem et al. baseline. Something similar happens also with the fair scheduling dataset in the appendix.
- It would make the results much stronger if there would be some indication that the techniques hold beyond the specific choice of $\alpha$-fair utility function. How are these techniques going to be generalized? It’s good that the authors give some examples, but is the proposed utility function capturing, at least qualitatively, how agents evaluate their reward?


**Questions:**

- Why examine $\alpha>1$? What does a utility function now mean?
- Why at Jain’s index is your policy better than opt offline? Should that be interpreted in some way?
- How do you think that your techniques could be generalized to other, more general concave utility functions? Can you give some extra reasoning of why you chose this particular form of utility function?

Minor comments:
-  In p.19 (p.7 of the appendix), erase the comment left there.
- Maybe you want to define somewhere the hitrate notion used in the experiments?



**Limitations:**

The authors adequately addressed the limitations.

---

> ### Author Rebuttal · Authors · 2023-08-06
>
> We thank the reviewer for their comments and questions, which we respond to below.
>
> [$\textbf{On the significance of the experimental results}$]: Although there is no major improvement in the average hit rate, the goal of our algorithm is to provide fairness (i.e., to ensure that the hit rates of all users are as close to each other as possible) without significantly sacrificing the average hit rate. To empirically show this fairness, we plot the minimum hit rate and Jain's Index (which is a standard notation of fairness that is maximized when all users have the same hit rate). It can be seen from the experiments that our algorithm has a higher Jain's Index and minimum hit rate than the baseline algorithms without sacrificing significantly in terms of average hit rate.
>
> [$\textbf{The alpha >1 regime}$] To intuitively understand the implication of the utility function for $\alpha > 1$, recall that the alpha-fairness function $\phi$ is proportional to $\sum_i \frac{1}{R_i(T)^{\alpha-1}}$. As $\alpha$ grows very large (i.e., much larger than $1$), this sum is dominated by $\frac{1}{\min_i R_i(T)^{\alpha-1}}$. Hence, maximizing this utility function for large $\alpha$ becomes nearly equivalent to maximizing the minimum cumulative reward. This is a natural fairness objective. Generally, increasing $\alpha$ from $0$ to $\infty$ yields a continuum of objective functions that increasingly weight fairness as compared to total utility.
>
> [$\textbf{Jain's Index}$] As described in line 315, Jain's Index is a metric that quantifies if the users are receiving a fair share of resources. In the context of fair caching, Jain's Index quantifies if the users are receiving similar hit rates, but it does not take into account whether the average hit rate is maximized. The Jain's Index of a policy can be high even when the average hit rate is low as long as the hit rates of individual users are close to each other. For example, in Figure 3, even though Jain's Index of our policy is higher than the optimal offline, the average hit rate is lower. Jain's Index does not capture the overall performance of a policy. The main reason we plot Jain's Index is to demonstrate that our policy (as well as the offline optimal) become "fairer" as the $\alpha$ parameter is increased. Comparing Jain's Index of different policies might not necessarily lead to any concrete conclusion.
>
> [$\textbf{Extension to General Concave Utility functions}$] The greedy policy can be extended straightforwardly for a general concave utility function $\phi$ by replacing the quantity $\frac{1}{R_i^\alpha}$ by $\phi' (R_i)$ in line 5 of Algorithm 1. However, with regards to the regret analysis, note that in Eq (10), we explicitly use the positive homogeneity property of the $\alpha$-fair utility function. Hence, we do not see an immediate way to extend the analysis for a general concave utility function. Nevertheless, in the special case, when the utility function $\phi$ is a sum of $\alpha$-fair utilities for different constant $\alpha$'s, the analysis goes through. It would be interesting to give even more general regret bounds for concave utilities.
>
> [$\textbf{Minor Comments}$] Thanks for pointing these out. We will make the necessary changes in the revised version.

---

> > ### Comment · Reviewer_q9U9 · 2023-08-18
> >
> > I thank the authors for their response.

---

### Official Review · Reviewer_DRz6 · 2023-07-09

**Soundness:** 3 good
**Presentation:** 3 good
**Contribution:** 3 good
**Rating:** 7
**Confidence:** 3

**Summary:**

This work studies an abstract fair resource allocation problem. It abstracts out problems such as cache and job scheduling. The notion of fairness considered is alpha-fairness (in the range of alpha between 0 and 1), which has been previously studied and is known to encapsulate many other fairness notions.

The paper provides sublinear approximate regret bounds for the abstract fair resource allocation problem through a clever reduction to online linear optimization, and analyzes it using prior work on online gradient descent in conjunction with a bootstrapping technique. The approximation constant is small (around 1.45) and they further show that the notion of approximate regret is necessary for this problem (i.e. that sublinear regret bounds require an approximation constant larger than 1). They also back up their theoretical results with experiments that compare to previous work and other standard benchmarks showing that their algorithms achieve good efficiency-fairness tradeoffs in comparison.


**Strengths:**

1- The approach of replacing the problem of interest with a surrogate problem that can be solved via more standard regret minimization techniques and relating the regrets of the two problems is a nice and novel contribution, as is the bootstrapping technique used to prove regret bounds for the surrogate problem.

2- It is nice that such results can be obtained without significantly restricting the adversary.


**Weaknesses:**

1- Assumption 2 seems reasonable for many problems, but Assumption 1 (that asks that demand vectors always be bounded away from 0) seems more restricting. Additionally, regret bounds seem to depend on the constants in these assumptions.

2- The paper discusses alpha-fairness for alpha between 0 and 1, but for alpha-fairness to encompass more fairness notions, larger alpha is relevant as well, and is unaddressed by their results

**Questions:**

1- In the proof of lemma 2, I’m not sure I understand what happens to the summation over i in equation 19. For example, when the regret is lower bounded by 1 for every user at every timestep (eqn 21), wouldn’t substituting in equation 19 give you O(sqrt(mT)) regret? Not sure if I’m missing something. Similar questions strike me in Case 1 and 2 in pages 18 and 19. I also don’t fully understand these cases and how they follow from equation 26, so would really appreciate more explanation.

2- Some more discussion on alpha-fairness and exposition on why it’s a good notion of fairness for these problems would be good.


**Limitations:**

Limitations adequately addressed.

---

> ### Author Rebuttal · Authors · 2023-08-06
>
> We thank the reviewer very much for their review and their questions, which we address below.
>
> [$\textbf{Assumption 1}$] We agree with the reviewer and believe that it might be possible to establish similar regret bounds without Assumption 1. Please keep in mind that, for many problems, e.g., the caching problem, where at least one file is requested per round, Assumption 1 is naturally satisfied with $\delta=1$.
> One possible way to remove Assumption 1 is to divide the entire horizon into two phases – a regular phase where Assumption 1 is satisfied and a special phase where the demand vectors are small and Assumption 1 is not satisfied. The regret analysis for the regular phase proceeds in the same way as given in the paper. Next, one could argue that since the demand vectors are small for the special phase, it contributes only a tiny amount to the overall regret. We reserve this extended analysis for future work.
>
> [$\textbf{Larger alpha}$]: Note that in our formulation, $R_i(T)$ is defined as the total cumulative reward of agent $i$, which grows linearly with time T. As we mentioned in Remark 1 (page 4, line 146), a sublinear regret bound becomes vacuous for any $\alpha >1$. Hence, we did not consider larger values of $\alpha$ in this paper. We agree though that establishing theoretical guarantees in the $\alpha > 1$ under some reasonable performance metric would be an interesting direction for future work.
>
> [$\textbf{Response to Q1}$]: For the simplicity of exposition, the $O(\cdot)$ notation hides dependencies on all parameters (including the number of users $m$) except the time horizon $T$. Also, please note that the variable $R_i(T)$ denotes the cumulative rewards accrued by $i$ th agent up to time T (not regret).
>
> [$\textbf{Intuitive explanation for bounding the L-Regret}$]: In a nutshell, the bootstrapping technique used in the proof of Lemma 2 is an algebraic exercise of simultaneously bounding the order of growth of the dependent sequences $\{R_i(t)\}$ and ${L-Regret}_{T}$ under the action of the proposed online policy. We initially start with a crude bound for both sequences and then successively refine these bounds by exploiting their interrelations.
>
> To be precise, we use Eq (19) for bounding the L-regret_T for any arbitrary round $T\geq 1$. From Eq (19), it follows that ${L-Regret}_{T}$ depends on the value of the cumulative reward sequence $\{R_i(t)\}$ – the larger the values of these variables, the tighter becomes the regret bound. Hence, the proof proceeds by establishing tight lower bounds for the cumulative reward sequence $\{R_i(t)\}$. This is established via Eq (26) which gives a lower bound to the cumulative reward accrued up to any round T.
>
> Let us now consider Case -I $(0\leq \alpha \leq 1/2)$. Since $L-Regret_T = O(\sqrt(T))$ from Eqn (22), Eqn (26) implies that $R_i(t)$ increases linearly with $t$, substituting this bound in Eq (19), we conclude that $L-Regret_T= O(\sqrt(\sum_t (\frac{1}{t^{2\alpha}})))$. We finally arrive at the stated regret bound upon summing this series. A similar explanation holds for Case II as well.
>
> [$\textbf{Response to Q2 - Motivation for alpha-fair utility function}$]: We thank the reviewer for the suggestion. In our response to Reviewer 1 above, we have provided a brief discussion on the Motivation for using the alpha-fair utility function. We will include this discussion in the revised version.

---

> > ### Comment · Reviewer_DRz6 · 2023-08-16
> >
> > I thank the authors for their response. My concerns are addressed.

---

### Official Review · Reviewer_jNhn · 2023-07-09

**Soundness:** 2 fair
**Presentation:** 2 fair
**Contribution:** 2 fair
**Rating:** 4
**Confidence:** 3

**Summary:**

The paper consider a fair resources allocation problem in the setting of unrestricted adversary, which is called a generic online fair resource allocation (NOFRA).  An OFA policy is proposed with reasonable theoretical guarantees.

**Strengths:**

The paper present an online fair allocation algorithm, which approximately maximizes the aggregate $\alpha$-fairness function. The lower bound of approximation factor is established.

**Weaknesses:**

I have several concerns in the problem formulation.

1. Why the initial condition $R_i(0) = 1$ instead of being 0? Maybe it is due to technical requirements?

2. To me, the motivation of using concave $\alpha$-fair utility function is not strong.

3. $c$-regret is adopted in the paper. To me, I do not quite understand why it is a good metric to evaluate policy. It seems it is merely for technical purposes?

4. Can author also highlight its technical novelty? Is Lemma 1 the most challenging?

5. The topic is about fair resource allocation. But the paper seems  to be loosely connected with fairness bandit. The definition of fairness here is also not clearly specified. The related literature should be included.

**Questions:**

See weakness part. I feel like the main content of this paper is to formulate the problem via using a different regret objective. It is very loosely related to usual fairness bandit. The writing should be improved.

**Limitations:**

No.

---

> ### Author Rebuttal · Authors · 2023-08-06
>
> We thank the reviewer for their comments. In the following, we address each of the comments in the same order.
>
> 1. [$\textbf{Justification for counting cumulative rewards from 1}$]: The initial value of $R_i(0)$ is set to $1$ instead of zero to make sure that the derivative of the $\alpha$-fair utility function $\phi(\cdot)$ remains finite at any round - a condition which is required by any online learning policy. Note that $\phi'(R_i)= 1/R_i^\alpha.$ Hence, the derivative of the utility function becomes infinite when $R_i=0.$ Thus, if we set the initial value of $R_i$ to one, and since R_i is a monotone nondecreasing function of time, this technical issue can be avoided. Another equivalent way to avoid this problem is to change the utility function to $\phi(x)= \frac{(1+x)^{(1-\alpha)}}{1-\alpha}$ and set $R_i(0)=0.$ However, we opted for the former option to avoid a more cumbersome expression for the utility function.
>
> 2. [$\textbf{Motivation for alpha-fair utility function}$]: We want to emphasize the $\alpha$-fair utility function is a standard utility function that enjoys several interesting technical properties and is heavily used in network resource allocation (see for example [1-2]). We also refer the reviewer to [3], which gives an axiomatic characterization of a fair utility function and shows that the alpha-fair utility function comes out naturally from the axioms. Other utility functions, e.g., proportional fair and min-max utilities, can be shown to be a limiting form of alpha-fair utility.
>
> 3. [$\textbf{The c-regret metric}$]: Regret is a standard metric to evaluate the performance of any online policy. However, as we show in Theorem 2, no online policy can achieve a sublinear regret for the problem that we study in this paper. The metric c-Regret generalizes the usual regret metric where the online policy achieves a sublinear regret against $(1/c)$-fraction of the utility achieved by a static adversary. The smaller the value of c, the stronger becomes the performance guarantee of the online policy. The notion of c-regret has also been extensively used in the online learning literature (see, e.g., the papers [4-6]). This is why we chose c-regret as a performance metric in this paper.
>
> 4. [$\textbf{Technical Novelty}$]: Both Lemma 1 and Lemma 2 are non-trivial results. Lemma 2 introduces a new successive refinement argument to obtain a tight regret bound.
>
> 5. [$\textbf{Related work on fair bandits}$]: In our current related work, there is a paragraph that covers related work in the context of multi-arm bandits (Lines 510-521). We will expand this with additional recent results.
>
> [1] Mo, Jeonghoon, and Jean Walrand. "Fair end-to-end window-based congestion control." IEEE/ACM Transactions on Networking 8.5 (2000): 556-567.
>
> [2] Li, Tian, Sanjabi, Maziar, Beirami, Ahmad, and Smith, Virginia. "Fair Resource Allocation in Federated Learning." International Conference on Learning Representations (ICLR). 2019.
>
> [3] T. Lan, D. Kao, M. Chiang and A. Sabharwal, "An Axiomatic Theory of Fairness in Network Resource Allocation," 2010 Proceedings IEEE INFOCOM, San Diego, CA, USA, 2010, pp. 1-9, doi: 10.1109/INFCOM.2010.5461911.
>
> [4] Azar, Yossi, Amos Fiat, and Federico Fusco. "An $\alpha $-regret analysis of Adversarial Bilateral Trade." Advances in Neural Information Processing Systems 35 (2022): 1685-1697.
>
> [5] Paria, Debjit, and Abhishek Sinha. "$\texttt {LeadCache} $: Regret-Optimal Caching in Networks." Advances in Neural Information Processing Systems 34 (2021): 4435-4447.
>
> [6] Emamjomeh-Zadeh, Ehsan, Chen-Yu Wei, Haipeng Luo, and David Kempe. "Adversarial online learning with changing action sets: Efficient algorithms with approximate regret bounds." In Algorithmic Learning Theory, pp. 599-618. PMLR, 2021.

---

### Decision · Program_Chairs · 2023-09-21

**Decision:**

Accept (poster)

**Comment:**

this paper looks at a standard problem of resource allocation (job scheduling, caching..) with the catch of adding a type of fairness constraint. This is an interesting contributions to the field that all the reviewers - and myself -, enjoyed. I therefore recommend acceptance